# Pessimism Meets Invariance: Provably Efficient Offline Mean-Field Multi-Agent RL[*]

**Minshuo Chen**[1]  **Yan Li**[1]  **Ethan Wang**[1]  **Zhuoran Yang**[2]  **Zhaoran Wang**[3]  **Tuo Zhao**[1]

[1]Georgia Tech    [2]University of California, Berkeley    [3]Northwestern University

{mchen393, tourzhao}@gatech.edu

## Abstract

Mean-Field Multi-Agent Reinforcement Learning (MF-MARL) is attractive in the applications involving a large population of homogeneous agents, as it exploits the permutation invariance of agents and avoids the curse of many agents. Most existing results only focus on online settings, in which agents can interact with the environment during training. In some applications such as social welfare optimization, however, the interaction during training can be prohibitive or even unethical in the societal systems. To bridge such a gap, we propose a SAFARI (peSsimistic meAn-Field vAlue iteRatIon) algorithm for off-line MF-MARL, which only requires a handful of pre-collected experience data. Theoretically, under a weak coverage assumption that the experience dataset contains enough information about the optimal policy, we prove that for an episodic mean-field MDP with a horizon $H$ and $N$ training trajectories, SAFARI attains a sub-optimality gap of $\mathcal{O}(H^2 d_{\text{eff}}/\sqrt{N})$, where $d_{\text{eff}}$ is the effective dimension of the function class for parameterizing the value function, but independent on the number of agents. Numerical experiments are provided.

## 1 Introduction

Significant progress has been made towards multi-agent reinforcement learning (MARL) for many prominent sequential decision making problems, such as social welfare optimization (Leibo et al., 2017), fleet control of autonomous vehicles (Shalev-Shwartz et al., 2016) and playing multiplayer online battle arena (MOBA) games (Berner et al., 2019). As the joint state and action space scales exponentially with the number of agents, however, MARL becomes computationally expensive. One remedy is the mean-field regime when an extremely large number of homogenous agents are involved, e.g., social welfare optimization. The effect of each agent on the overall multi-agent system can become infinitesimal, and therefore all agents can be considered interchangeable/indistinguishable (Yang et al., 2018; Carmona et al., 2019; Li et al., 2021). Accordingly, the interaction among agents can be captured by some mean-field quantity such as the empirical distribution of states, and therefore each agent only needs to find the best response to the so-called "mean-field state", which avoids the curse of many agents.

Most existing results on mean-field MARL (MF-MARL) are for the online setting (Yang et al., 2018; Zhang et al., 2019), where the agents can interact with the environment during training. However, such interaction during training can be prohibitive for some important applications (Leibo et al., 2017; Mandel et al., 2014; Jaques et al., 2019; Levine et al., 2020). Taking social welfare optimization as an example, repeatedly conducting social experiments on human being can be unaffordable or even unethical in the societal systems. Therefore, we can only consider the offline settings, i.e., we learn the optimal policy based on some pre-collected experience data (Levine et al., 2020). Unfortunately,

---

[*]Extension to online setting is provided in a longer technical report version, which is available upon request.

35th Conference on Neural Information Processing Systems (NeurIPS 2021).

existing offline reinforcement learning (RL) algorithms and theories all focus on the single agent settings, and no algorithms and theories have been developed for MARL under the offline settings, regardless of the mean-field regime or not.

To bridge such a critical gap, we propose the first pessimistic algorithm – named SAFARI (peSsimistic meAn-Field vAlue iteRatIon) for mean-field MARL, which can provably achieve sample efficiency under the offline setting. Our proposed algorithm contains two important components: (1) To incorporate the permutation invariance of the homogenous agents, we adopt a RKHS (Reproducing Kernel Hilbert Space) mean-embedding approach for approximating value functions, which avoids the exponential blowup of the agents' state and action spaces; (2) We develop an uncertainty quantifier, and integrate it into the value iteration procedure as the penalty function. Such a penalty function can effectively screen the "spuriously correlated trajectories", i.e., which possibly happen to appear in the experience data, but are actually unrelated to the optimal policy, but by chance induce large cumulative rewards and hence may potentially mislead the learned policy.

Theoretically, we establish a data-dependent upper bound on the suboptimality of SAFARI for MF-MARL without the stringent assumptions on the sufficient coverage of the experience data (e.g., finite concentrability coefficients (Chen and Jiang, 2019) or uniformly lower bounded densities of visitation measure (Yin et al., 2020)). More specifically, we only assume that the experience data of $N$ training trajectories contains enough information about the optimal policy. Then we prove that for an episodic MF-MARL problem with a horizon $H$, SAFARI attains a sub-optimality gap of $\mathcal{O}(H^2 d_{\text{eff}}/\sqrt{N})$, where $d_{\text{eff}}$ is the effective dimension of the function class (RKHS) for parameterizing the value function and independent on the number of agents. In addition to the offline settings, our SAFARI algorithm can also be extended to MF-MARL under the online setting (OMPPO algorithm), which is of independent interest. Details are provided in a longer technical report version, which is available upon request.

The rest of this paper is organized as follows: Section 2 reviews related work on mean-field multi-agent reinforcement learning and offline reinforcement learning for the single agent settings; Section 3 introduces our problem setup of the mean-field MARL regime; Section 4 introduces our proposed SAFARI algorithm; Section 5 establishes the theoretical guarantees for SAFARI; Section 6 presents numerical experiments on the multi-agent particle cooperative navigation scenario; Section 7 draws a brief conclusion.

## 2 Related Work

• **Mean-Field MARL**. Existing literature has proposed various mean-field approximation approaches to model the population behavior of the agents for MARL with a large number, even infinitely many homogenous agents. Yang et al. (2017) investigate a mean-field game with deterministic linear state transitions, and reformulate it as a mean-field MDP, where the mean-field state lies in finite-dimensional probability simplex. Yang et al. (2018) propose a mean-field approximation approach over actions, which approximates the interaction between any given agent and the population by the interaction between the agent's action and the averaged actions of its neighboring agents. Such an averaging approach over the local actions, however, is only applicable when a sparse graph over agents is given, which requires extensive prior knowledge. Carmona et al. (2019) investigate a mean-field MDP from the perspective of mean-field control. As the mean-field state lies in a probability simplex and continuous in nature, they propose to discretize the joint state-action space such that conventional RL algorithms can be applied. Wang et al. (2020) investigate a mean-field MDP motivated by permutation invariance. They require a central controller managing the actions of all the agents, and therefore is restricted to handling the curse of many agents from the exponential blowup of joint state space. More recently, Li et al. (2021) investigate a similar mean-field MDP, which allows agents to make their own local actions without resorting to a centralized controller. All these methods focus on the online settings. In comparison, our proposed SAFARI algorithm and theory focus on the offline settings.

• **RL for Mean-Field Game.** Our work is also related to the literature that studies RL methods for mean-field games (Huang et al., 2003; Lasry and Lions, 2006a,b; Huang et al., 2007). Such a game can be viewed as the infinite-agent limit of general-sum Markov game with homogeneous agents, and the aggregated effect of the other agents is also summarized as a mean-field state. In contrast to mean-field MARL, the solution concept of mean-field game is the Nash equilibrium,

which corresponds to a pair of a local policy $\pi^*$ of the representative agent and a mean-field state $d^*$ satisfying the following two properties: (i) when the mean-field state is set to $d^*$, $\pi^*$ is the optimal policy of the representative agent; and (ii) when all agents adopt $\pi^*$, the resulting mean-field state is $d^*$. Recently, there are many recent works developing RL methods for solving mean-field games. See, e.g., Guo et al. (2019, 2020b,a); Fu et al. (2019); Anahtarcı et al. (2019); Anahtarci et al. (2020); Anahtarcı et al. (2020); Perrin et al. (2020); Elie et al. (2020); uz Zaman et al. (2020); Cui and Koeppl (2021) and the references therein. Most of these methods adopts a double-loop structure, where the inner loop finds the optimal local policy given the current mean field state and the outer loop updates the mean-field states. Moreover, these works often assume the data distribution is well-explored with either a generative model (Azar et al., 2012) or bounded concentrability coefficients (Munos, 2007). Our mean-field MARL problem is similar to the inner-loop problem of finding the optimal local policy in mean-field games. In contrast to these existing works, our algorithm and theory can be applied to datasets that are possibly not well-explored. Moreover, as mean-field MARL and mean-field games are different models, our work is not directly comparable to these works.

• **Offline Single-Agent RL**. Our work is also closely related to the literature on offline single-agent RL, which often focuses on either policy evaluation or policy optimization. In particular, in policy evaluation, the goal is to estimate the value function of a target policy, whereas in policy optimization, we aims to learn the optimal policy, which can be achieved via estimating the optimal value function. For both these tasks, in the offline setting, due to the lack of continuing exploration (Szepesvári, 2010), the distribution shift (Levine et al., 2020) is a fundamental challenge. That is, the trajectories in the dataset and those induced by the target policy or the optimal policy might have diverse distributions. Such a challenge is further exacerbated when function approximators are adopted to represent the desired value functions. To overcome such a challenge, most of the existing theoretical works imposes certain well-exploration assumptions on the dataset. Some of commonly made assumptions include uniformly lower bounded visitation measure of the behavior policy, uniformly upper bounded importance sampling ratio, and bounded concentrability coefficients. See, e.g., Antos et al. (2007, 2008); Munos and Szepesvári (2008); Farahmand et al. (2010, 2016); Scherrer et al. (2015); Jiang and Li (2016); Thomas and Brunskill (2016); Farajtabar et al. (2018); Liu et al. (2018); Xie et al. (2019); Nachum et al. (2019a,b); Tang et al. (2019); Zhang et al. (2020b); Chen and Jiang (2019); Kallus and Uehara (2019, 2020); Jiang and Huang (2020); Uehara et al. (2020); Duan et al. (2020); Yin and Wang (2020); Yin et al. (2020); Nachum and Dai (2020); Yang et al. (2020a); Fu et al. (2020b); Fan et al. (2020); Xie and Jiang (2020a,b); Liao et al. (2020); Zhang et al. (2020a); Ren et al. (2021) and the references therein.

However, in practice, such assumptions on the dataset often fail to hold (Fujimoto et al., 2019; Agarwal et al., 2020; Fu et al., 2020a; Gulcehre et al., 2020). In light of this, there is a line of recent works that proposes various pessimism-based offline single-agent RL algorithms with empirical evidence or theoretical guarantees (Yu et al., 2020; Kidambi et al., 2020; Kumar et al., 2020; Liu et al., 2020b; Buckman et al., 2020; Jin et al., 2020b; Xiao et al., 2021). In particular, Liu et al. (2020b) propose a regularized variant of fitted Q-iteration (Antos et al., 2007, 2008; Munos and Szepesvári, 2008), which is shown to attain the optimal policy within a restricted policy class without assuming the dataset is well-explored. Moreover, with an arbitrary dataset, Buckman et al. (2020); Jin et al. (2020b); Xiao et al. (2021) identify the critical role of pessimism in achieving offline sample efficiency. Among these works, our work is particularly related to Jin et al. (2020b), which develops a pessimistic variant of the value iteration algorithm with finite-dimensional linear function approximation. In comparison, our SAFARI algorithm extends such an algorithm to mean-field MARL and we propose to employ RKHS mean embedding for handling the difference between finite-agent empirical mean-field state and its infinite-agent counterpart. Moreover, our algorithm and analysis involve infinite-dimensional RKHS, which strictly generalizes those in Jin et al. (2020b).

**Notation**: Given a space $\mathcal{X}$, we denote $\mathcal{M}(\mathcal{X})$ as the collection of probability distributions supported on $\mathcal{X}$. Let $u, v, w \in \mathcal{H}$ be elements in a Hilbert space, we denote $\langle u, v \rangle$ as the inner product, and $u \otimes v$ as the outer product satisfying $(u \otimes v)w = u \langle v, w \rangle$. For a scalar $a$, we denote $\{a\}^+ = \max\{0, a\}$. We use $\mathcal{O}(\cdot)$ to hide absolute constants and log factors.

# 3 Mean-Field Multi-Agent RL

We consider a Multi-Agent Reinforcement Learning (MARL) problem with $m + 1$ agents and time horizon $H$. For the $i$-th agent (also known as the Representative Agent (RA)), at step $h$, we denote $s_{i,h} \in \mathcal{S}$ and $a_{i,h} \in \mathcal{A}$ as its state and action, respectively. We assume $\mathcal{S}$ and $\mathcal{A}$ are compact.

Different from single agent RL problem, the transition kernel, reward function, and policy of a representative agent in MARL depend not only on its individual state, but the states of $m$ other agents. Furthermore, we assume that the interaction of the representative agent to the other agents is permutation invariant, i.e., the influence of all the other agents is modeled using the empirical distribution of states $\widehat{d}_{s,h} = \frac{1}{m} \sum_{j \neq i}^{m} \delta_{s_{j,h}} \in \mathcal{M}(\mathcal{S})$. To this end, we define the transition kernel $p_h : \mathcal{S} \times \mathcal{M}(\mathcal{S}) \times \mathcal{A} \mapsto \mathcal{M}(\mathcal{S})$, the (deterministic) reward function $r_h : \mathcal{S} \times \mathcal{M}(\mathcal{S}) \times \mathcal{A} \mapsto \mathbb{R}$, and the policy $\pi_h : \mathcal{S} \times \mathcal{M}(\mathcal{S}) \mapsto \mathcal{M}(\mathcal{A})$ all depending on a "meta state" denoted as $\widehat{\omega}_h = (s_{i,h}, \widehat{d}_{s,h}) \in \mathcal{S} \times \mathcal{M}(\mathcal{S})$. For simplicity, we denote $\Omega = \mathcal{S} \times \mathcal{M}(\mathcal{S})$ as the meta state space.

**Remark 1.** The empirical distribution of states $\widehat{d}_{s,h}$ is naturally permutation invariant and evolves according to the transition kernel $p_h$ and policy $\pi_h$. To see this, suppose each agent takes the same policy $\pi_h$ at step $h$. Then at step $h + 1$, the state $s_{h+1,j}$ of the $j$-th agent is sampled from the distribution $p_h(\cdot \mid s_{h,j} \times \widehat{d}_{s,h}, a_{h,j})$, where $a_{h,j}$ is determined by policy $\pi_h(\cdot \mid s_{h,j} \times \widehat{d}_{s,h})$. Collecting $m$ states $s_{h+1,j}$ for $j \neq i$ induces the empirical distribution of states $\widehat{d}_{s,h+1}$.

We now define several important notions in MARL. Given a policy $\pi$, the value function $V_h^{\pi} : \Omega \mapsto \mathbb{R}$ at step $h \leq H$ for a representative agent is

$$V_h^{\pi}(\omega) = \mathbb{E}_{\pi} \left[ \sum_{i=h}^{H} r_i(\omega_i, a_i) \mid \omega_h = \omega \right], \tag{1}$$

where $\mathbb{E}_{\pi}$ denotes the expectation over the randomness in trajectories induced by policy $\pi$. The action-value function (Q-function) $Q_h^{\pi} : \Omega \times \mathcal{A} \mapsto \mathbb{R}$ is defined as

$$Q_h^{\pi}(\omega, a) = \mathbb{E}_{\pi} \left[ \sum_{i=h}^{H} r_i(\omega_i, a_i) \mid \omega_h = \omega, a_h = a \right].$$

By definition, $V_h^{\pi}$ and $Q_h^{\pi}$ are related via $V_h^{\pi}(\omega) = \int_{\mathcal{A}} Q_h^{\pi}(\omega, a) \pi(a|\omega) da \overset{\triangle}{=} \langle Q_h^{\pi}, \pi \rangle_{\mathcal{A}}$. Next, we define the Bellman operator and conditional transition operator. At each step $h \leq H$, the Bellman operator denoted as $\mathbb{B}_h$ is

$$(\mathbb{B}_h g)(\omega, a) = \mathbb{E}\left[ r_h(\omega_h, a_h) + g(\omega_{h+1}) \mid \omega_h = \omega, a_h = a \right] \tag{2}$$
$$= r_h(\omega, a) + (\mathbb{P}_h g)(\omega, a),$$

where $g$ is a function defined on $\Omega$, and $\mathbb{P}_h$ is referred to as the conditional transition operator.

**Mean-Field MARL**  As the number of agents goes to infinity, the empirical distribution of states $\widehat{d}_s$ converges to a (continuous) limit $d_s$. Then the mean-field MARL problem for a representative agent is defined as a tuple $(\Omega, \mathcal{A}, H, P, r)$, where $\Omega$ and $\mathcal{A}$ are the meta state space and action space, respectively, $H$ is the horizon, $P = \{p_h\}_{h=1}^{H} : \Omega \times \mathcal{A} \mapsto \mathcal{M}(\mathcal{S})$ is the transition kernel, and $r = \{r_h\}_{h=1}^{H}$ is the reward function defined on $\Omega \times \mathcal{A}$. Following Remark 1, the transition of $d_s$ is also determined by $P = \{p_h\}_{h=1}^{H}$.

To tackle the infinite-dimensional joint distribution of states, we embed the meta state-action space $\Omega \times \mathcal{A}$ into a reproducing kernel Hilbert space (RKHS). Specifically, denote $\Xi = \mathcal{S} \times \mathcal{S} \times \mathcal{A}$ and let $K : \Xi \times \Xi \mapsto \mathbb{R}$ be a symmetric positive kernel. The corresponding feature mapping of kernel $k$ is denoted as $\psi$, which verifies $\langle \psi(\cdot), \psi(\cdot) \rangle = K(\cdot, \cdot)$ can be infinite dimensional. For any $(\omega, a) \in \Omega \times \mathcal{A}$, we define mean embedding as

$$\mu(\omega, a) = \mathbb{E}_{s' \sim d_s}[\psi(s, s', a)]. \tag{3}$$

Based on the embedding, we parameterize the reward $r_h$ and Markov transition $p_h$ as linear functionals of $\mu(\omega, a)$ in RKHS $\mathcal{H}_K$ induced by kernel $K$, i.e.,

$$r_h(\omega, a) = \langle \mu(\omega, a), \theta_h \rangle, \quad p_h(\omega' \mid \omega, a) = \langle \mu(\omega, a), v_h(\omega') \rangle, \tag{4}$$

where $\theta_h, v_h$ are understood as "weights" and have bounded Hilbert norm (see Assumption 3). Such a parameterization encodes a rich family of functions, once the kernel is universal (Wang et al., 2020). By the definition of $Q$-function and value function, we can show that the Bellman operator can also be parameterized in $\mathcal{H}_K$.

**Proposition 1.** Suppose the reward function $r_h$ and the transition kernel $p_h$ is parameterized in $\mathcal{H}_K$ by (4) for $h = 1, \ldots, H$. Then for any $g : \Omega \mapsto \mathbb{R}$, the Bellman operator $(\mathbb{B}_h g)$ and conditional transition operator $(\mathbb{P}_h g)$ defined in (2) can be written as

$$(\mathbb{B}_h g)(\omega, a) = \langle \mu(\omega, a), w_g \rangle, \quad (\mathbb{P}_h g)(\omega, a) = \langle \mu(\omega, a), w_g + \theta_h \rangle,$$

where $w_g$ depends on the function $g$.

The proof is provided in Appendix C.1, which follows from pure algebraic manipulation. From the perspective of policy learning in mean-field MARL, Proposition 1 motivates us to estimate the Bellman operator $\mathbb{B}_h$ in $\mathcal{H}_K$, and then optimize the estimated $Q$-function to obtain a policy. We introduce the detailed learning procedure in Section 4 (Algorithm 1).

# 4 Offline Pessimistic Value Iteration

In this section, we introduce our dataset and learning algorithm. We collect multiple trajectories of a representative agent in a mean-field MARL problem. Here the mean-field state distribution $d_s$ is prohibitive to trace. Instead, we only independently observe the states of a finite number of agents. Accordingly, the batched dataset $\mathcal{D}_{N,H}$ consists of $N$ trajectories of length $H$, within which the $n$-th sequence is $\tau_n = \left\{ (s_h^n \in \mathcal{S}^{m+1}, a_h^n \in \mathcal{A}, r_h^n \in \mathbb{R}) \right\}_{h=1}^H$. Without loss of generality, we assume $s_{h,0}$ is the state of the representative agent, and the reward function is bounded by 1, i.e., $|r_h(\omega, a)| \leq 1$ for any $\omega \in \Omega, a \in \mathcal{A}$. The collected trajectories are generated by some unknown behavior policy.

Recall $\widehat{d}_{s_h^n} = \frac{1}{m} \sum_{j=1}^m s_{h,j}^n$ is the empirical state distribution induced by $s_h^n$. (We slightly alter the notation to emphasize the empirical distribution is generated by the collection of $m$ states $s_{h,1:m}^n$, while in the previous context, we use a general purpose notation $\widehat{d}_{s,h}$.) We denote $\widehat{\omega}_h^n = s_{h,0}^n \times \widehat{d}_{s_h^n}$, and compute the empirical mean embedding of $(\widehat{\omega}_h^n, a_h^n)$ as

$$\mu(\widehat{\omega}_h^n, a_h^n) = \mathbb{E}_{s' \sim \widehat{d}_{s_h^n}} [\psi(s_{h,0}^n, s', a_h^n)] = \frac{1}{m} \sum_{j=1}^m \psi(s_{h,0}^n, s_{h,j}^n, a_h^n).$$

Under mild conditions, the empirical mean embedding $\mu(\widehat{\omega}_h^n, a_h^n)$ concentrates around the infinite agent mean embedding $\mu(\omega_h^n, a_h^n)$ defined in (3), where $\omega_h^n$ is the infinite agent meta state. See a detailed error quantification in Lemma 3.

**Pessimistic Value Iteration** Our goal is to learn an optimal policy to be deployed for all the agents based on the experience data of the representative agent. The idea is to estimate the $Q$-function at each time step in the RKHS $\mathcal{H}_K$, and then optimize the $Q$-function to obtain an optimal policy. In more detail, at step $h \leq H$, we estimate Bellman operator by optimizing the empirical mean squared Bellman error

$$(\widehat{\mathbb{B}}_h \widehat{V}_{h+1}) = \underset{f}{\arg\min} \sum_{n=1}^N \left( f(\mu(\widehat{\omega}_h^n, a_h^n)) - r_h^n - \widehat{V}_{h+1}(\widehat{\omega}_{h+1}^n) \right)^2 + \lambda \|f\|_{\mathcal{H}}^2, \tag{5}$$

where $\lambda \geq 1$ controls the regularization strength, $\widehat{V}$ is the estimated value function, and $\|\cdot\|_{\mathcal{H}}$ denotes the Hilbert norm.

The solution to (5) can be written in a closed form. For notational simplicity, we define

$$K((\omega, a), \cdot) = \mathbb{E}_{s' \sim d_s}[K((s, s', a), \cdot)] \quad \text{with} \quad \omega = s \times d_s.$$

Then we denote the Gram matrix $K_h \in \mathbb{R}^{N \times N}$ as

$$[K_h]_{\ell, \ell'} = K((\widehat{\omega}_h^\ell, a_h^\ell), (\widehat{\omega}_h^{\ell'}, a_h^{\ell'})) \triangleq \mathbb{E}_{s_1 \sim \widehat{d}_h^\ell, s_2 \sim \widehat{d}_h^{\ell'}} \langle \psi(s_{h,0}^\ell, s_1, a_h^\ell), \psi(s_{h,0}^{\ell'}, s_2, a_h^{\ell'}) \rangle$$

for $\ell, \ell' = 1, \ldots, N$. Meanwhile, for any $(\omega, a)$, we denote feature vector $\phi_h(\omega, a) = \left[ K((\widehat{\omega}_h^1, a_h^1), (\omega, a)), \ldots, K((\widehat{\omega}_h^N, a_h^N), (\omega, a)) \right]^\top \in \mathbb{R}^N$. Then the estimated Bellman operator

$\widehat{\mathbb{B}}_h \widehat{V}_{h+1}$ can be written as

$$(\widehat{\mathbb{B}}_h \widehat{V}_{h+1})(\omega, a) = \phi_h(\omega, a)^\top \widehat{\alpha}_h$$
$$\text{with} \quad \widehat{\alpha}_h = (K_h + \lambda I)^{-1}[r_h^1 + \widehat{V}_{h+1}(\widehat{\omega}_{h+1}^1), \ldots, r_h^N + \widehat{V}_{h+1}(\widehat{\omega}_{h+1}^N)]^\top, \tag{6}$$

We summarize the proposed SAFARI algorithm in Algorithm 1.

---

**Algorithm 1** Pessimistic Mean-Field Value Iteration (SAFARI)

---

**Input**: Dataset $\mathcal{D}_{N,H}$, coefficient $\beta$, regularization coefficient $\lambda$.
**Initialize**: Set $\widehat{V}_{H+1} = 0$.
**for** $h = H, H-1, \ldots, 1$ **do**
    Compute $\Lambda_h = K_h + \lambda I$.
    Estimate $\widetilde{Q}_h(\omega, a) \triangleq (\widehat{\mathbb{B}}_h \widehat{V}_{h+1})(\omega, a) = \phi_h(\omega, a)^\top \widehat{\alpha}_h$ as in (6).
    Set $\Gamma_h(\omega, a) = \beta \cdot \lambda^{-1/2} \left( K((\omega, a), (\omega, a)) - \phi_h(\omega, a)^\top \Lambda_h^{-1} \phi_h(\omega, a) \right)^{1/2}$.
    Let $\widehat{Q}_h(\omega, a) = \min\{\widetilde{Q}_h(\omega, a) - \Gamma_h(\omega, a), H - h + 1\}^+$.
    Optimal policy $\widehat{\pi}_h = \text{argmax}_\pi \langle \widehat{Q}_h(\omega, \cdot), \pi(\cdot \mid \omega) \rangle_{\mathcal{A}}$.
    Set $\widehat{V}_h(\omega) = \langle \widehat{Q}_h(\omega, \cdot), \widehat{\pi}_h(\cdot \mid \omega) \rangle_{\mathcal{A}}$.
**end for**
**Output**: Estimated $Q$-function $\widehat{Q}_h$, value function $\widehat{V}_h$, and optimal policy $\widehat{\pi}_h$ for $h = 1, \ldots, H$.

---

The quantity $\Gamma_h$ quantifies the uncertainty in estimating the Bellman operator $\mathbb{B}_h \widehat{V}_{h+1}$ using kernel ridge regression. We subtract $\Gamma_h$ for estimating the Bellman operator to account for the spurious correlation in the experience data (see Technical Overview following Theorem 1 for a detailed explanation). We truncate $\widehat{Q}_h$ at $H - h + 1$, since the reward function is bounded by 1.

## 5 Suboptimality of Policy Learned by SAFARI

We investigate the performance of the optimal policy $\widehat{\pi}$ learned by Algorithm 1. Before we proceed, we state the following assumptions.

**Assumption 1** (Boundedness of Kernel). Kernel $K(\cdot, \cdot)$ is bounded, i.e., without loss of generality, we assume $\sup_{\xi \in \Xi} |K(\xi, \xi)| \leq 1$.

By Cauchy-Schwarz inequality, Assumption 1 implies for any $\xi_1, \xi_2 \in \Xi$, $K(\xi_1, \xi_2) \leq \sqrt{K(\xi_1, \xi_1)K(\xi_2, \xi_2)} \leq 1$. Such an assumption holds for a rich family of commonly used kernels, e.g., RBF kernel and Laplacian kernel, and is a standard assumption in literature (Caponnetto and De Vito, 2007; Muandet et al., 2012).

The second assumption characterizes the spectrum of kernel $K$. We first introduce the integral operator induced by kernel $K$. Let $f : \Xi \mapsto \mathbb{R}$ be a square-integrable function. Then we define the integral operator $\mathcal{T}_K$ as

$$(\mathcal{T}_K f)(\xi) = \int K(\xi, x) f(x) dx \quad \text{for} \quad \xi \in \Xi.$$

By Mercer's theorem (Hearst et al., 1998), $\mathcal{T}_K$ has corresponding positive eigenvalues $\sigma_i$ and eigenfunctions $\nu_i$. Then the kernel $K$ admits a decomposition

$$K(\xi_1, \xi_2) = \sum_{i=1}^\infty \sigma_i \nu_i(\xi_1) \nu_i(\xi_2).$$

**Assumption 2** (Spectrum of Kernel). The eigenvalue $\sigma_i$ satisfies one of the following three conditions:

1. *(Finite Spectrum)*. There exists a positive integer $\gamma$, such that $\sigma_i = 0$ for all $i > \gamma$.

2. *(Exponential Decay)*. There exist positive constants $C_1, C_2$ and exponent $\gamma > 0$ such that $\sigma_i \leq C_1 \exp(-C_2 i^\gamma)$.

3. *(Polynomial Decay)*. There exists a positive constant $C$ and exponent $\gamma \geq 3 + \mathcal{O}(\frac{1}{d})$ such that $\sigma_i \leq Ci^{-\gamma}$, where $d$ is the dimension of $\mathcal{S} \times \mathcal{S} \times \mathcal{A}$.

Furthermore, in *(Exponential Decay)* and *(Polynomial Decay)*, we assume the eigenfunction $\nu_i$ is uniformly bounded, i.e., $\sup_i \|\nu_i\|_\infty \leq 1$.

As we will show in our theory, the decay rate of the spectrum significantly influences the performance of the proposed SAFARI algorithm. We give examples to better interpret the three categories above. In *(Finite Spectrum)* case, by (4), the reward function and transition kernel is a linear function of a finite dimensional feature map. Such a parameterization is satisfied by linear MDP (Yang and Wang, 2019; Jin et al., 2020a). In *(Exponential Decay)* and *(Polynomial Decay)* cases, the feature map is infinite dimensional. For example, RBF kernel belongs to *(Exponential Decay)* case, while Laplacian kernel and neural tangent kernel belong to *(Polynomial Decay)* case. We assume $\gamma \geq 3 + \mathcal{O}(1/d)$ in *(Polynomial Decay)* for technical simplicity, yet it is not restrictive: Laplacian kernel and neural tangent kernel both have a polynomial decay rate of $\gamma = d$ (Bietti and Bach, 2020).

The last assumption imposes some regularity on the reward function and transition probabilities.

**Assumption 3** (Boundedness)**.** The weights $\theta_h$ and $v_h$ in reward function $r_h$ and Markov transition kernel $p_h$ are bounded for any $h = 1, \ldots, H$, respectively, i.e., $\|\theta_h\|_{\mathcal{H}} \leq 1$ and $\int_\Omega \|v_h(x)\|_{\mathcal{H}} \, dx \leq \sqrt{d_{\mathrm{eff}}}$, where $d_{\mathrm{eff}} = \sup_{K_h} \log \det(I + K_h/\lambda)$ is the effective dimension of $\mathcal{H}_K$ with supremum over all Gram matrix $K_h \in \mathbb{R}^{N \times N}$.

The effective dimension describes the complexity of $\mathcal{H}_K$ for parameterizing the MDP (Yang et al., 2020b), whose scale is closely related to the spectrum of kernel $K$. In the special case of $K$ having a $\gamma$-finite spectrum as in Assumption 2, we have $d_{\mathrm{eff}} = \mathcal{O}(\gamma)$, which resembles the dimensionality of a finite dimensional Euclidean space.

We measure the pointwise suboptimality of the learned policy $\widehat{\pi}$. We define the global optimal policy by the recursion,

$$\pi_h^* = \operatorname*{argmax}_\pi \ \langle Q_h^*, \pi \rangle_{\mathcal{A}}, \quad \text{with} \quad Q_h^* = \mathbb{B}_h V_{h+1}^*, V_h^* = \langle Q_h^*, \pi_h^* \rangle_{\mathcal{A}}, \text{ and } V_{H+1}^* = 0.$$

Then the suboptimality of $\widehat{\pi}$ is given as

$$\mathtt{SubOpt}(\widehat{\pi}; \omega) = V_1^{\pi^*}(\omega) - V_1^{\widehat{\pi}}(\omega).$$

Our main result is provided in the following theorem, which upper bounds $\mathtt{SubOpt}(\widehat{\pi}; \omega)$.

**Theorem 1.** Suppose Assumption $1 - 3$ hold. For any $\delta \in (0, 1)$, let $\widehat{\pi}_h$ be the policy returned by Algorithm 1 with

$$m \geq \log(2/\delta), \quad \lambda = 1, \quad \beta = \begin{cases} c \max\{d, \gamma\} H \sqrt{\log(\max\{d, \gamma\} H N/\delta)} & \textit{(Finite Spectrum)} \\ cH \sqrt{d \left(\log(HN/\delta)\right)^{1+2/\gamma}} & \textit{(Exponential Decay)} \\ cN^{\frac{d+1}{d+\gamma}} H \sqrt{d \log(HN/\delta)} & \textit{(Polynomial Decay)} \end{cases},$$

where $d$ is the dimension of $\Xi = \mathcal{S} \times \mathcal{S} \times \mathcal{A}$ and $c$ is some constant depending on $C, C_1, C_2$ and Lebesgue measure of $\Xi$. Then for any meta state $\omega$, with probability at least $1 - \delta$ over the randomness of the dataset $\mathcal{D}_{N,H}$, we have

$$\mathtt{SubOpt}(\widehat{\pi}; \omega) \leq 2 \sum_{h=1}^H \mathbb{E}_{\pi^*}[\Gamma_h(\omega_h, a_h) \mid \omega_1 = \omega].$$

Theorem 1 indicates that the suboptimality of learned policy depends on the uncertainty quantifier $\Gamma_h$. The scale of $\Gamma_h$ depends on how well the collected data explore the state-action space. Moreover, from a Bayesian learning perspective, $\Gamma_h$ measures the eliminated uncertainty in estimating the Bellman operator given dataset $\mathcal{D}_{N,H}$ (Jin et al., 2020b). To better understand the convergence of $\mathtt{SubOpt}$, we specialize Theorem 1 under a weak data coverage assumption.

**Assumption 4** (Weak Coverage)**.** Suppose the dataset is collected under some behavior policy $\bar{\pi}$ such that there exists a constant $c_{\mathrm{min}} > 0$ satisfying

$$\inf_{\|f\|_{\mathcal{H}}=1} \langle f, \mathbb{E}_{\bar{\pi}}[\mu(\omega_h, a_h) \otimes \mu(\omega_h, a_h)] f \rangle \geq c_{\mathrm{min}} \quad \text{for any} \quad h = 1, \ldots, H.$$

Recall that $\mu$ is the mean embedding in $\mathcal{H}_K$.

Assumption 4 says that the operator $\mathbb{E}_{\bar{\pi}}[\mu(\omega_h, a_h) \otimes \mu(\omega_h, a_h)]$ is positive definite. Intuitively, this requires that the collected data relatively well spread over the state-action space. We present the following Corollary providing a concrete convergence rate of SubOpt.

**Corollary 1.** Under the setting in Theorem 1, we additionally assume Assumption 4 holds. Then for $N \geq \Omega(\log(d_{\text{eff}} H/\delta))$ sufficiently large, with probability $1 - \delta$, we have

$$\texttt{SubOpt}(\widehat{\pi}; \omega) = \mathcal{O}\left(H^2 d_{\text{eff}} \sqrt{\frac{\log(d_{\text{eff}} H N/\delta)}{N}}\right).$$

Here $d_{\text{eff}}$ is the effective dimension of RKHS $\mathcal{H}_K$, which takes value

$$d_{\text{eff}} = \begin{cases} \max\{d, \gamma\} \log N & \textit{(Finite Spectrum)} \\ d(\log N)^{1+1/\gamma} & \textit{(Exponential Decay)} \\ dN^{\frac{d+1}{d+\gamma}} \log N & \textit{(Polynomial Decay)} \end{cases}.$$

**Impact of Kernel Spectrum** The spectrum of kernel $K$ significantly influences the performance of the learned policy. In *(Finite Spectrum)* case, the effective dimension scales linearly with dimension $d$ and $\gamma$, and SubOpt converges at a rate of $\mathcal{O}(H^2 \max\{d, \gamma\}/\sqrt{N})$, which recovers the result of Corollary 4.5 in Jin et al. (2020b) on linear MDP. In *(Exponential Decay)* case, the convergence rate is $\mathcal{O}(H^2 d(\log N)^{1+1/\gamma}/\sqrt{N})$, which is similar to *(Finite Spectrum)* case with additional logarithmic dependence on $N$. However, in *(Polynomial Decay)* case, the convergence rate is considerably slower, and relies heavily on the decay rate $\gamma$. Consider, for instance, Laplacian kernel and NTK, whose spectrum decays with $\gamma = d$. Then SubOpt converges at a rate of $\mathcal{O}(H^2 dN^{-\frac{1}{2d}} \log N)$, which suffers from the curse of dimensionality without further assumptions on data.

**No Curse of Many Agents** The convergence of SubOpt does not suffer from the curse of many agents. In particular, both Theorem 1 and Corollary 1 only impose a mild requirement on the number $m$ of neighboring agents to be sampled. This is due to the permutation invariance in mean-field MARL, since the interactive influence of neighboring agents are captured by the distribution of states.

**Technical Overview** We briefly discuss the proof of Theorem 1 and Corollary 1. The full proof is deferred to Appendix A and B. We first decompose SubOpt into three terms (see Lemma 1):

$$\texttt{SubOpt}(\pi; \omega) = \mathcal{E}_1 + \mathcal{E}_2 + \mathcal{E}_3.$$

Here $\mathcal{E}_1 = \sum_{h=1}^{H} \mathbb{E}_{\widehat{\pi}}[\widehat{Q}_h(\omega_h, a_h) - (\mathbb{B}_h \widehat{V}_{h+1})(\omega_h, a_h) \mid \omega_1 = \omega]$ reflects the uncertainty in estimating the Bellman operator. Note that the evaluating trajectory is generated by the learned policy $\widehat{\pi}$, which has spurious correlation with the estimated Bellman operator; $\mathcal{E}_2 = \sum_{h=1}^{H} \mathbb{E}_{\pi^*}[(\mathbb{B}_h \widehat{V}_{h+1})(\omega_h, a_h) - \widehat{Q}_h(\omega_h, a_h) \mid \omega_1 = \omega]$ is the estimation error of Bellman operator again, yet it is evaluated by a trajectory generated by $\pi^*$. Compared to $\mathcal{E}_1$, $\mathcal{E}_2$ does not suffer from the spurious correlation between the learned policy and the estimated Bellman operator. Lastly, $\mathcal{E}_3 = \sum_{h=1}^{H} \mathbb{E}_\pi[\langle \widehat{Q}_h(\omega_h, \cdot), \pi_h^*(\cdot \mid \omega_h) - \widehat{\pi}_h(\cdot \mid \omega_h) \rangle_{\mathcal{A}} \mid \omega_1 = \omega]$ is the optimization error. By the optimality of $\widehat{\pi}$, we immediately have $\mathcal{E}_3 \leq 0$.

In order to tackle $\mathcal{E}_1$ and $\mathcal{E}_2$, we properly choose $\Gamma_h$ so that the event $E = \{|\mathbb{B}_h \widehat{V}_{h+1} - \widehat{\mathbb{B}}_h \widehat{V}_{h+1}| \leq \Gamma_h\}$ happens with high probability. In fact, $\Gamma_h$ is understood as the uncertainty quantifier of estimating $\mathbb{B}_h \widehat{V}_{h+1}$ with high confidence $1 - \delta$. Then we can show $\mathcal{E}_1 \leq 0$ conditioned on event $E$, meanwhile $\mathcal{E}_2 \leq 2 \sum_{h=1}^{H} \mathbb{E}_{\pi^*}[\Gamma_h(\omega_h, a_h) \mid \omega_1 = \omega]$. To this end, we reduce the upper bound of SubOpt to bounding the uncertainty quantifier $\Gamma_h$, which allows us to leverage statistical tools. In particular, $\Gamma_h$ consists of two types of statistical error: 1) covariate concentration error on mean embedding, i.e., finite agent empirical embedding $\mu(\widehat{\omega}_h^n, a_h^n)$ concentration error with respect to population counterpart $\mu(\omega_h^n, a_h^n)$; 2) regression error in Bellman operator estimation. We bound 1) by concentration of empirical means in Hilbert spaces (see Lemma 3). In bounding 2), we exploit the closed form solution of kernel ridge regression and concentration of self-normalizing processes (see Lemma 5).

## 6 Numerical Experiment

We perform experiments on the multi-agent particle environment (MPE, Lowe et al. (2017)), a popular benchmark used in prior work (Mordatch and Abbeel, 2018; Liu et al., 2020a). Here, we consider the

*cooperative navigation* scenario, where $N$ agents must spread to cooperatively cover $N$ landmarks across the map. Each agent is able to observe information about the $k$ closest landmarks and agents, and receives a global reward $r = -\sum_{i=1}^{N} \min_{j \in [N]} \|y_i - x_j\|_2$, where $x_i \in \mathbb{R}^2$ and $y_i \in \mathbb{R}^2$ are agent and landmark positions, respectively. Implementation of all environments follows from the official codebase of Liu et al. (2020a). See hyperparameter choices and more details in Appendix E. Sample code is also available at `https://github.com/wange011/offline-pessimistic`.

**Data Generation** To receive optimal reward on *cooperative navigation*, individual agent must learn to coordinate their behaviors to each cover a different landmark. As a result, we generate data for the offline setting by training a MARL policy and collecting experience data after convergence. We use *counterfactual multi-agent policy gradients* (COMA) to address the problem of *credit assignment* by learning a joint critic that marginalizes out an individual agent's action with a counterfactual baseline (Foerster et al., 2018). This, in turn, allows the agent-level policies to learn sufficient coordination by evaluating their individual impact on the team reward. Both the policy and critic networks are implemented as traditional MLPs, with $64$ and $512$ nodes in a single hidden layer, respectively, and we

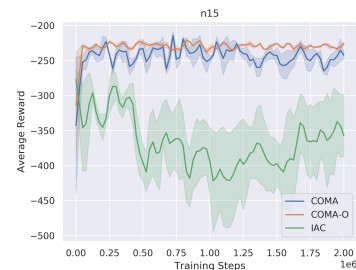

Figure 1: Training reward on the 15 agent environment.

use parameter sharing for policy networks. To sanity check the performance of COMA, we train the *individual actor-critic* (IAC) algorithm (Konda and Tsitsiklis, 2000), which applies the policy gradient to train independent actor-critics. Given the lack of an in-built coordination mechanism, IAC is expected to perform suboptimally on multi-agent settings.

As all agents take the same action in the mean-field MARL formulation, COMA produces experiences by selecting the action that corresponds with the plurality vote (mode) of individual agent policy outputs. However, to demonstrate that this does not greatly inhibit convergence behavior, we train IAC and the original COMA implementation, labeled COMA-O, without this restriction. As demonstrated in Figure 1, with error bar computed over 3 independent random seeds, COMA-O performs the best. It is worth noting that COMA receives slightly lower rewards yet still performing significantly better than IAC with the same number of learnable parameters.

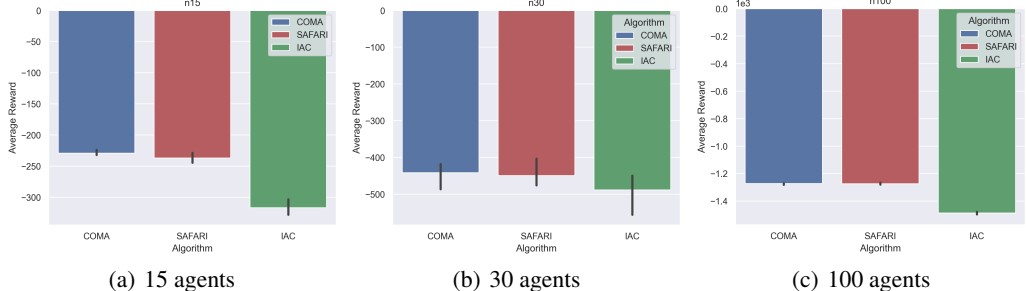

(a) 15 agents         (b) 30 agents         (c) 100 agents

Figure 2: Average reward after training. COMA and IAC are evaluated off loaded pre-trained models.

In Figure 2, we implement our SAFARI algorithm with varying number of agents on $n = 500$ sample episodes of experience data. We evaluate the performance over a horizon $H = 50$ on 3 different random seeds. We observe that SAFARI is able to perform comparably to COMA in settings with $m = 15$, $30$, and $100$ agents. Due to mean-field permutation invariance, we see that the performance gap between SAFARI and COMA does not widen as the number of agents increases, a behavior that is normally expected given the exponential growth of the joint state-action space.

## 7 Conclusion

This paper proposes a SAFARI (Pessimistic Mean-Field Value Iteration) algorithm in offline mean-field MARL. We prove a suboptimality bound $\mathcal{O}(H^2 d_{\text{eff}}/\sqrt{N})$, and provide concrete rate of convergence under a weak data coverage assumption. The suboptimality bound is free of the curse of many agents due to the permutation invariance in mean-field formulation. We also extend to the online setting in a longer technical report version.

## Acknowledgment

Zhaoran Wang acknowledges National Science Foundation (Awards 2048075, 2008827, 2015568, 1934931), Simons Institute (Theory of Reinforcement Learning), Amazon, J.P. Morgan, and Two Sigma for their supports. Zhuoran Yang acknowledges Simons Institute (Theory of Reinforcement Learning).

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
