## A   Proof of Theorem 1 — Offline Pessimistic Policy Learning

Throughout the proofs, we adopt the following convention on inner product and outer product in Hilbert space $\mathcal{H}_K$. Let $V = [v_1, \ldots, v_d]^\top$ and $U = [u_1, \ldots, u_d]$ be collections of elements in $\mathcal{H}_K$ ($v_i, u_j \in \mathcal{H}$). For any $w \in \mathcal{H}_K$, we denote $\langle V, w \rangle = [\langle v_1, w \rangle, \ldots, \langle v_d, w \rangle]^\top \in \mathbb{R}^d$, meanwhile, $\langle V^\top, w \rangle = \langle V, w \rangle^\top \in \mathbb{R}^d$ is a row vector. We also denote $\langle V^\top, U \rangle = \sum_{i=1}^d \langle v_i, u_i \rangle \in \mathbb{R}$, while $\langle V, U^\top \rangle \in \mathbb{R}^{d \times d}$ is a matrix. For outer product, we similarly denote $V^\top \otimes U = \sum_{i=1}^d v_i \otimes u_i$ and $V \otimes U^\top = [v_i \otimes u_j]_{i,j}$ as operators. Such a convention coincides with the standard vector algebra in finite dimensional spaces.

*Proof.* The full proof consists of four steps. In each step, we require several technical lemmas, whose proofs are deferred to Appendix C.

**Step 1: Suboptimality Decomposition**. We decompose SubOpt into three terms.

**Lemma 1.** Given a policy $\pi = \{\pi_h\}_{h=1}^H$ and $Q$-function $\{Q_h\}_{h=1}^H$ with $V_h = \langle Q_h, \pi_h \rangle_{\mathcal{A}}$, for any meta state $\omega$, SubOpt can be decomposed into three terms,

$$
\texttt{SubOpt}(\pi; \omega) = \underbrace{\sum_{h=1}^H \mathbb{E}_\pi \left[ Q_h(\omega_h, a_h) - (\mathbb{B}_h V_{h+1})(\omega_h, a_h) \mid \omega_1 = \omega \right]}_{\mathcal{E}_1}
$$
$$
+ \underbrace{\sum_{h=1}^H \mathbb{E}_{\pi^*} \left[ (\mathbb{B}_h V_{h+1})(\omega_h, a_h) - Q_h(\omega_h, a_h) \mid \omega_1 = \omega \right]}_{\mathcal{E}_2} \tag{7}
$$
$$
+ \underbrace{\sum_{h=1}^H \mathbb{E}_\pi \left[ \langle Q_h(\omega_h, \cdot), \pi_h^*(\cdot \mid \omega_h) - \pi_h(\cdot \mid \omega_h) \rangle_{\mathcal{A}} \mid \omega_1 = \omega \right]}_{\mathcal{E}_3}.
$$

The proof is provided in Appendix C.2. We instantiate $\pi$, $Q_h$, and $V_h$ in Lemma 1 to $\widehat{\pi}_h$, $\widehat{Q}_h$, and $\widehat{V}_h$ returned by Algorithm 1. The optimality of $\widehat{\pi}_h$, i.e., $\widehat{\pi}_h = \operatorname{argmax}_\pi \langle \widehat{Q}_h, \pi_h \rangle_{\mathcal{A}}$ implies that the third term $\mathcal{E}_3$ in (7) is non-positive. Therefore, $\texttt{SubOpt}(\widehat{\pi}; \omega)$ admits the upper bound

$$
\texttt{SubOpt}(\widehat{\pi}; \omega) \leq \mathcal{E}_1 + \mathcal{E}_2. \tag{8}
$$

**Step 2: Pessimism Correction and Simplified Suboptimality Upper Bound**. We further simplify (8) by assuming the following concentration condition:

$$
\left| (\mathbb{B}_h \widehat{V}_{h+1})(\omega, a) - (\widehat{\mathbb{B}}_h \widehat{V}_{h+1})(\omega, a) \right| \leq \Gamma_h(\omega, a) \quad \text{for any} \quad (\omega, a) \in \Xi. \tag{9}
$$

We will show in **Step 4** that choosing $\beta$ and $\lambda = 1$ as in Theorem 1, $\Gamma_h$ computed in Algorithm 1 verifies condition (9) with probability at least $1 - \delta$. Conditioned on (9), we can show term $\mathcal{E}_1$ is negative and term $\mathcal{E}_2$ is bounded by $2\Gamma_h$.

**Lemma 2.** In the setup of Theorem 1, let $\widehat{Q}_h$ and $\Gamma_h$ be computed as in Algorithm 1. Conditioned on (9), for any $(\omega, a)$, the following sandwich inequality holds true,

$$
0 \leq (\mathbb{B}_h \widehat{V}_{h+1})(\omega, a) - \widehat{Q}_h(\omega, a) \leq 2\Gamma_h(\omega, a), \quad \text{for} \quad h = 1, \ldots, H.
$$

The proof is provided in Appendix C.3. Lemma 2 immediately implies that term $\mathcal{E}_1$ in (7) is non-positive, and $\mathcal{E}_2$ is upper bounded by $2\Gamma_h$. As a result, we simplify (8) as

$$\texttt{SubOpt}(\widehat{\pi}; \omega) \leq 2 \sum_{h=1}^{H} \mathbb{E}_{\pi^*} \left[ \Gamma_h(\omega_h, a_h) \mid \omega_1 = \omega \right]. \tag{10}$$

**Step 3: Establishing Condition** (9). Recall that we require $\Gamma_h(\omega, a)$ satisfying

$$\left| (\mathbb{B}_h \widehat{V}_{h+1})(\omega, a) - (\widehat{\mathbb{B}}_h \widehat{V}_{h+1})(\omega, a) \right| \leq \Gamma_h(\omega, a) \quad \text{for any} \quad (\omega, a) \in \Xi,$$

with probability at least $1 - \delta$. It suffices to characterize the concentration of $\widehat{\mathbb{B}}_h \widehat{V}_{h+1}$ to $\mathbb{B}_h \widehat{V}_{h+1}$. By Proposition 1, we can write $\mathbb{B}_h \widehat{V}_{h+1} = \langle \mu(\omega, a), \alpha_h \rangle$ for some weight $\alpha_h$. We also denote $\widehat{r}_h^n = r_h(\widehat{\omega}_h^n, a_h^n)$. We then decompose $(\mathbb{B}_h \widehat{V}_{h+1})(\omega, a) - (\widehat{\mathbb{B}}_h \widehat{V}_{h+1})(\omega, a)$ into three terms,

$$(\mathbb{B}_h \widehat{V}_{h+1})(\omega, a) - (\widehat{\mathbb{B}}_h \widehat{V}_{h+1})(\omega, a)$$
$$= \langle \mu(\omega, a), \alpha_h \rangle - \phi_h(\omega, a)^\top \widehat{\alpha}_h$$
$$= \langle \mu(\omega, a), \alpha_h \rangle - \phi_h(\omega, a)^\top \Lambda_h^{-1} \left[ \widehat{r}_h^1 + \widehat{V}_{h+1}(\widehat{\omega}_{h+1}^1), \ldots, \widehat{r}_h^N + \widehat{V}_{h+1}(\widehat{\omega}_{h+1}^N) \right]^\top$$
$$+ \underbrace{\phi_h(\omega, a)^\top \Lambda_h^{-1} \left[ \widehat{r}_h^1 - r_h^1, \ldots, \widehat{r}_h^N - r_h^N \right]^\top}_{(A)}$$
$$= (A) + \underbrace{\langle \mu(\omega, a), \alpha_h \rangle - \phi_h(\omega, a)^\top \Lambda_h^{-1} \left[ (\mathbb{B}_h \widehat{V}_{h+1})(\widehat{\omega}_n^1, a_h^1), \ldots, (\mathbb{B}_h \widehat{V}_{h+1})(\widehat{\omega}_n^N, a_h^N) \right]^\top}_{(B)}$$
$$- \underbrace{\phi_h(\omega, a)^\top \Lambda_h^{-1} \left[ \widehat{r}_h^1 + \widehat{V}_{h+1}(\widehat{\omega}_{h+1}^1) - \mathbb{B}_h \widehat{V}_{h+1}(\widehat{\omega}_h^1, a_h^1), \ldots, \widehat{r}_h^N + \widehat{V}_{h+1}(\widehat{\omega}_{h+1}^N) - \mathbb{B}_h \widehat{V}_{h+1}(\widehat{\omega}_h^N, a_h^N) \right]^\top}_{(C)}.$$

Consequently, we have

$$\left| (\mathbb{B}_h \widehat{V}_{h+1})(\omega, a) - (\widehat{\mathbb{B}}_h \widehat{V}_{h+1})(\omega, a) \right| \leq |(A)| + |(B)| + |(C)|.$$

Intuitively, term $(A)$ measures the error induced by the empirical estimation of the mean-field distribution of states. Term $(B)$ corresponds to the bias of kernel ridge regression, and Term $(C)$ is the statistical error. We tackle these terms separately in the sequel.

● **Bounding Term** $(A)$. We show reward $\widehat{r}_h^n$ concentrates around $r_h^n$, by establishing the concentration of empirical mean embedding to its population counterpart.

**Lemma 3.** Let $\widehat{\omega}_m$ be the empirical mean embedding corresponding to $m$ agents i.i.d. sampled from infinite-agent state distribution. Given any $\delta_A > 0$, with probability at least $1 - \delta_A$, for any $a \in \mathcal{A}$, we have

$$\| \mu(\widehat{\omega}_m, a) - \mu(\omega, a) \|_{\mathcal{H}_K} \leq \sqrt{\frac{2}{m}} + \sqrt{\frac{2 \log(1/\delta_A)}{m}}.$$

The proof is provided in Appendix C.4. Combining Assumption 3 and Lemma 3, with probability $1 - \delta_A$, it holds

$$\sup_n |\widehat{r}_h^n - r_h^n| \leq \|\theta_h\|_{\mathcal{H}_K} \left( \sqrt{\frac{2}{m}} + \sqrt{\frac{2 \log(1/\delta_A)}{m}} \right) \leq \left( \sqrt{\frac{2}{m}} + \sqrt{\frac{2 \log(1/\delta_A)}{m}} \right). \tag{11}$$

We are now ready to prove the following upper bound on $(A)$.

**Lemma 4.** Suppose Assumption 1 and 2 hold. With probability $1 - \delta_A$, it holds

$$|(A)| \leq 2 \left( \sqrt{\frac{1}{m}} + \sqrt{\frac{\log(1/\delta_A)}{m}} \right) \sqrt{\log \det(I + K_h/\lambda)} \left\| \Sigma_h^{-1/2} \mu(\omega, a) \right\|_{\mathcal{H}}.$$

The proof is provided in Appendix C.5. We note that $\log \det(I + K_h/\lambda)$ is known as the effective dimension of an RKHS (Yang et al., 2020b). The scale of $\log \det(I + K_h/\lambda)$ closely ties to the spectrum of kernel $K$. See Lemma 10 for an upper bound on $\log \det(I + K_h/\lambda)$. It is enough to set $\delta_A = \delta/2$, which yields

$$|(A)| \leq 2\left(\sqrt{\frac{1}{m}} + \sqrt{\frac{\log(2/\delta)}{m}}\right)\sqrt{\log \det(I + K_h/\lambda)}\left\|\Sigma_h^{-1/2}\mu(\omega, a)\right\|_{\mathcal{H}} \qquad (12)$$

with probability $1 - \delta/2$.

• **Bounding Term** $(B)$. We derive a useful decomposition of the mean embedding $\mu$ to simplify term $(B)$. We let $\Phi_h = [\mu_1, \ldots, \mu_N]^\top$ for any fixed collection of $\mu_1, \ldots, \mu_N$. Then we define the regularized covariance operator as

$$\Sigma_h = \lambda I_{\mathcal{H}_K} + \Phi_h^\top \otimes \Phi_h,$$

where $I_{\mathcal{H}_K}$ is the identity operator on $\mathcal{H}_K$, and $\Phi_h^\top \otimes \Phi_h = \sum_{n=1}^N \mu_1 \otimes \mu_N$. The operator $\Sigma_h$ has eigenvalues lower bounded by $\lambda$. Therefore, its inverse operator $\Sigma_h^{-1}$ is well-defined. Now we check the identity

$$\begin{aligned}
\Sigma_h^{-1}\Phi^\top &= \left(\lambda I_{\mathcal{H}_K} + \Phi_h^\top \otimes \Phi_h\right)^{-1}\Phi_h^\top \\
&\overset{(i)}{=} \Phi_h^\top\left(\lambda I + \left\langle \Phi_h, \Phi_h^\top\right\rangle\right)^{-1} \\
&= \Phi_h^\top\left(\lambda I + K_h\right)^{-1},
\end{aligned} \qquad (13)$$

where $\left\langle \Phi_h, \Phi_h^\top\right\rangle = [\langle\mu_\ell, \mu_{\ell'}\rangle]_{\ell, \ell'} \in \mathbb{R}^{N \times N}$, and equality $(i)$ follows from

$$\left(\lambda I_{\mathcal{H}_K} + \Phi_h^\top \otimes \Phi_h\right)\Phi_h^\top = \lambda\Phi_h^\top + (\Phi_h^\top \otimes \Phi_h)\Phi_h^\top = \lambda\Phi_h^\top + \Phi_h^\top\left\langle \Phi_h, \Phi_h^\top\right\rangle = \Phi_h^\top\left(\lambda I + \left\langle \Phi_h, \Phi_h^\top\right\rangle\right)$$

which implies $\Phi_h^\top\left(\lambda I + \left\langle \Phi_h, \Phi_h^\top\right\rangle\right)^{-1} = \left(\lambda I_{\mathcal{H}_K} + \Phi_h^\top \otimes \Phi_h\right)^{-1}\Phi_h^\top$. We are ready to decompose mean embedding $\mu$ as

$$\begin{aligned}
\mu(\cdot) &= \Sigma_h^{-1}\Sigma_h\mu(\cdot) \\
&= \Sigma_h^{-1}\left(\lambda I_{\mathcal{H}_K} + \Phi_h^\top \otimes \Phi_h\right)\mu(\cdot) \\
&= \lambda\Sigma_h^{-1}\mu(\cdot) + \Sigma_h^{-1}\Phi_h^\top \otimes \Phi_h\mu(\cdot) \\
&\overset{(i)}{=} \lambda\Sigma_h^{-1}\mu(\cdot) + \left(\Phi_h^\top\left(\lambda I + K_h\right)^{-1} \otimes \Phi_h\right)\mu(\cdot) \\
&\overset{(ii)}{=} \lambda\Sigma_h^{-1}\mu(\cdot) + \Phi_h^\top\left(\lambda I + K_h\right)^{-1}\left\langle \Phi_h, \mu(\cdot)\right\rangle \\
&= \lambda\Sigma_h^{-1}\mu(\cdot) + \Phi_h^\top\left(\lambda I + K_h\right)^{-1}\phi_h(\cdot) \\
&= \lambda\Sigma_h^{-1}\mu(\cdot) + \Phi_h^\top\Lambda_h^{-1}\phi_h(\cdot),
\end{aligned} \qquad (14)$$

where step $(i)$ follows from (13), and step $(ii)$ uses the definition of outer product. We use (14) to simplify $(B)$ and derive an upper bound. We overload $\Phi_h$ by replacing fixed collection $\mu_1, \ldots, \mu_N$ with $\mu(\widehat{\omega}_h^1, a_h^1), \ldots, \mu(\widehat{\omega}_h^N, a_h^N)$. By substituting $\mathbb{B}_h\widehat{V}_{h+1}(\omega, a) = \langle\mu(\omega, a), \alpha_h\rangle$ into $(B)$, we have

$$\begin{aligned}
(B) &= \langle\mu(\omega, a), \alpha_h\rangle - \phi_h(\omega, a)^\top\Lambda_h^{-1}\left[\langle\mu(\widehat{\omega}_h^1, a_h^1), \alpha_h\rangle, \ldots, \langle\mu(\widehat{\omega}_h^N, a_h^N), \alpha_h\rangle\right]^\top \\
&\overset{(i)}{=} \left\langle\lambda\Sigma_h^{-1}\mu(\omega, a) + \Phi_h^\top\Lambda_h^{-1}\phi_h(\omega, a), \alpha_h\right\rangle \\
&\quad - \phi_h(\omega, a)^\top\Lambda_h^{-1}\left[\langle\mu(\widehat{\omega}_h^1, a_h^1), \alpha_h\rangle, \ldots, \langle\mu(\widehat{\omega}_h^N, a_h^N), \alpha_h\rangle\right]^\top \\
&= \lambda\left\langle\Sigma_h^{-1}\mu(\omega, a), \alpha_h\right\rangle.
\end{aligned}$$

By Cauchy-Schwarz inequality, we have

$$(B) \leq \lambda\left\|\Sigma_h^{-1/2}\mu(\omega, a)\right\|_{\mathcal{H}}\left\|\Sigma_h^{-1/2}\alpha_h\right\|_{\mathcal{H}} \overset{(i)}{\leq} \sqrt{\lambda}\|\alpha_h\|_{\mathcal{H}}\left\|\Sigma_h^{-1/2}\mu(\omega, a)\right\|_{\mathcal{H}},$$

where inequality $(i)$ follows from the operator norm of $\Sigma_h^{-1}$ being upper bounded by $\lambda^{-1}$. To finish bounding term $(B)$, we derive an upper bound on $\|\alpha_h\|_{\mathcal{H}}$. By Proposition 1, we have

$$\alpha_h = \int_\Omega \widehat{V}_{h+1}(x)v_h(x)dx + \theta_h.$$

The estimated value function satisfies $|\widehat{V}_{h+1}| \leq H - h$. Therefore, by the triangle inequality, we deduce

$$\|\alpha_h\|_{\mathcal{H}} \leq \left\| \int_\Omega \widehat{V}_{h+1}(x) v_h(x) dx \right\|_{\mathcal{H}} + \|\theta_h\|_{\mathcal{H}}$$

$$\leq (H - h) \int_\Omega \|v_h(x)\|_{\mathcal{H}} \, dx + \|\theta_h\|_{\mathcal{H}}$$

$$\overset{(i)}{\leq} (H - h)\sqrt{d_{\text{eff}}} + 1$$

$$\leq H\sqrt{d_{\text{eff}}},$$

where inequality $(i)$ holds due to Assumption 3. Consequently, we derive

$$|(B)| \leq \sqrt{\lambda} H \sqrt{d_{\text{eff}}} \left\| \Sigma_h^{-1/2} \mu(\omega, a) \right\|_{\mathcal{H}}. \tag{15}$$

- **Bounding Term** $(C)$. We use (13) to write

$$\phi_h(\omega, a)^\top \Lambda_h^{-1} = \left\langle \mu(\omega, a), \Phi_h^\top \right\rangle \Lambda_h^{-1} = \left\langle \mu(\omega, a), \Phi_h^\top \Lambda_h^{-1} \right\rangle = \left\langle \mu(\omega, a), \Sigma_h^{-1} \Phi_h^\top \right\rangle. \tag{16}$$

Denote $\Delta^n(\widehat{V}_{h+1}) = \widehat{r}_h^n + \widehat{V}_{h+1}(\widehat{\omega}_{h+1}^n) - \mathbb{B}_h \widehat{V}_{h+1}(\widehat{\omega}_h^n, a_h^n)$. Then term $(C)$ can be rewrite as

$$(C) = \left\langle \mu(\omega, a), \Sigma_h^{-1} \Phi_h^\top \right\rangle \left[ \Delta^1(\widehat{V}_{h+1}), \ldots, \Delta^n(\widehat{V}_{h+1}) \right]^\top$$

$$= \left\langle \mu(\omega, a), \Sigma_h^{-1} \sum_{n=1}^N \mu(\widehat{\omega}_h^n, a_h^n) \Delta^n(\widehat{V}_{h+1}) \right\rangle$$

$$\overset{(i)}{\leq} \left\| \Sigma_h^{-1/2} \mu(\omega, a) \right\|_{\mathcal{H}} \underbrace{\left\| \Sigma_h^{-1/2} \sum_{n=1}^N \mu(\widehat{\omega}_h^n, a_h^n) \Delta^n(\widehat{V}_{h+1}) \right\|_{\mathcal{H}}}_{(\star)},$$

where $(i)$ invokes Cauchy-Schwarz inequality. We construct a space of functions that contains $\widehat{V}_h$ to decouple the dependence between the data $\mathcal{D}_{N,H}$ and $\widehat{V}_h$ in $(\star)$. Specifically, we define $\mathcal{V}_h$ consisting of functions in the form of

$$\mathcal{V}_h(R, B, \lambda) =$$

$$\left\{ V_h(\omega) : V_h(\omega) = \max_{a \in \mathcal{A}} \left[ \min \left\{ \langle \mu(\omega, a), \theta \rangle - \beta \sqrt{\langle \mu(\omega, a), \Sigma \cdot \mu(\omega, a) \rangle}, H - h + 1 \right\}^+ \right], \right.$$

$$\left. \|\theta\|_{\mathcal{H}} \leq R, \beta \in [0, B], \lambda^{-1} I_{\mathcal{H}_K} \succeq \Sigma \succeq 0 \right\}. \tag{17}$$

Note that when taking $\Sigma = \Sigma_h^{-1}$ in $\beta \sqrt{\langle \mu(\omega, a), \Sigma \cdot \mu(\omega, a) \rangle}$, it becomes an equivalent form of $\Gamma_h(\omega, a)$. To see this, we take inner product on both sides of (14) with $\psi$, and derive

$$K((\omega, a), (\omega, a)) = \langle \mu(\omega, a), \mu(\omega, a) \rangle$$

$$= \lambda \left\langle \mu(\omega, a), \Sigma_h^{-1} \mu(\omega, a) \right\rangle + \left\langle \mu(\omega, a), \Phi^\top \right\rangle \Lambda_h^{-1} \phi_h(\omega, a)$$

$$= \lambda \left\langle \mu(\omega, a), \Sigma_h^{-1} \mu(\omega, a) \right\rangle + \phi_h(\omega, a)^\top \Lambda_h^{-1} \phi_h(\omega, a).$$

By rearranging terms, we deduce

$$\lambda \left\langle \mu(\omega, a), \Sigma_h^{-1} \mu(\omega, a) \right\rangle = K((\omega, a), (\omega, a)) - \phi_h(\omega, a)^\top \Lambda_h^{-1} \phi_h(\omega, a),$$

and further $\Gamma_h(\omega, a) = \beta \sqrt{\left\langle \mu(\omega, a), \Sigma_h^{-1} \mu(\omega, a) \right\rangle}$. As a result, we have $\widehat{V}_h \in \mathcal{V}_h(R, B, \lambda)$ for properly chosen $B$ and $R$, which are determined in **Step 4**.

We discretize $\mathcal{V}_{h+1}(R, B, \lambda)$ with respect to the $\ell_\infty$ norm, and find the closest element to replace $\widehat{V}_{h+1}$. In more detail, for any $\epsilon > 0$, we denote $\{V_{h+1,j}\}_{j=1}^{\mathcal{N}(\epsilon, \mathcal{V}_{h+1}(R, B, \lambda), \|\cdot\|_\infty)}$ as an $\epsilon$-covering of

$\mathcal{V}_{h+1}(R, B, \lambda)$, where $\mathcal{N}(\epsilon, \mathcal{V}_{h+1}(R, B, \lambda), \|\cdot\|_\infty)$ is known as the covering number. By definition, there exists an index $J$, such that $\left\|\widehat{V}_{h+1} - V_{h+1,J}\right\|_\infty \leq \epsilon$. By the triangle inequality, we have

$$
\begin{aligned}
(\star) &= \left\|\Sigma_h^{-1/2} \sum_{n=1}^N \mu(\widehat{\omega}_h^n, a_h^n) \left(\Delta^n(\widehat{V}_{h+1}) - \Delta^n(V_{h+1,J}) + \Delta^n(V_{h+1,J})\right)\right\|_{\mathcal{H}} \\
&\leq \left\|\Sigma_h^{-1/2} \sum_{n=1}^N \mu(\widehat{\omega}_h^n, a_h^n) \left(\Delta^n(\widehat{V}_{h+1}) - \Delta^n(V_{h+1,J})\right)\right\|_{\mathcal{H}} + \left\|\Sigma_h^{-1/2} \sum_{n=1}^N \mu(\widehat{\omega}_h^n, a_h^n) \Delta^n(V_{h+1,J})\right\|_{\mathcal{H}}.
\end{aligned}
$$

The first term above can be bounded by

$$
\begin{aligned}
&\left\|\Sigma_h^{-1/2} \sum_{n=1}^N \mu(\widehat{\omega}_h^n, a_h^n) \left(\Delta^n(\widehat{V}_{h+1}) - \Delta^n(V_{h+1,J})\right)\right\|_{\mathcal{H}} \\
&\stackrel{(i)}{=} \left\|\Sigma_h^{-1/2} \sum_{n=1}^N \mu(\widehat{\omega}_h^n, a_h^n) \left(\widehat{V}_{h+1}(\widehat{\omega}_{h+1}^n) - \mathbb{B}_h \widehat{V}_{h+1}(\widehat{\omega}_h^n, a_h^n) - \widehat{V}_{h+1,J}(\widehat{\omega}_{h+1}^n) + \mathbb{B}_h \widehat{V}_{h+1,J}(\widehat{\omega}_h^n, a_h^n)\right)\right\|_{\mathcal{H}} \\
&\stackrel{(ii)}{\leq} 2\epsilon \left\|\Sigma_h^{-1/2} \sum_{n=1}^N \mu(\widehat{\omega}_h^n, a_h^n)\right\|_{\mathcal{H}} \\
&\stackrel{(iii)}{\leq} 2\epsilon \lambda^{-1/2} N \max_{\ell, \ell'} \left|K((\widehat{\omega}_h^\ell, a_h^\ell), (\widehat{\omega}_h^{\ell'}, a_h^{\ell'}))\right| \\
&\leq 2\epsilon \lambda^{-1/2} N,
\end{aligned}
$$

where equality $(i)$ uses the definition of $\Delta^n$, inequality $(ii)$ follows from the definition of $\widehat{V}_{h+1,J}$, and inequality $(iii)$ holds since $\Sigma_h \succeq \lambda I_{\mathcal{H}}$. Consequently, we bound $(\star)$ as

$$
(\star) \leq \sup_{j \leq \mathcal{N}(\epsilon, \mathcal{V}_{h+1}(R, B, \lambda), \|\cdot\|_\infty)} \left\|\Sigma_h^{-1/2} \sum_{n=1}^N \mu(\widehat{\omega}_h^n, a_h^n) \Delta^n(V_{h+1,j})\right\|_{\mathcal{H}} + 2\epsilon \lambda^{-1/2} N. \tag{18}
$$

A crucial observation is that $V_{h+1,j}$ no longer coupled with the data $\mathcal{D}_{N,H}$, which allows us to derive uniform concentration on the first term above. We need the following lemma.

**Lemma 5** (Restatement of Lemma B.2 in Jin et al. (2020b)). For any $h \leq H$, let $V_{h+1}$ be a given value function. Then, for any $\delta_C \in (0, 1)$, we have

$$
\mathbb{P}\left(\left\|\Sigma_h^{-1/2} \sum_{n=1}^N \mu(\widehat{\omega}_h^n, a_h^n) \Delta^n(V_{h+1})\right\|_{\mathcal{H}}^2 > H^2 \left(2 \log \frac{1}{\delta_C} + \log \det(\lambda I + K_h)\right)\right) \leq \delta_C.
$$

The proof is provided in Appendix C.6. Taking union bound over the covering of $\mathcal{V}_{h+1}$, we immediately have

$$
\begin{aligned}
\mathbb{P}\left(\sup_j \left\|\Sigma_h^{-1/2} \sum_{n=1}^N \mu(\widehat{\omega}_h^n, a_h^n) \Delta^n(V_{h+1,j})\right\|_{\mathcal{H}}^2 > H^2 \left(2 \log \frac{1}{\delta_C} + \log \det(\lambda I + K_h)\right)\right) \\
\leq \delta_C \mathcal{N}(\epsilon, \mathcal{V}_{h+1}(R, B, \lambda), \|\cdot\|_\infty).
\end{aligned}
$$

We choose $\delta_C = \mathcal{N}^{-1}(\epsilon, \mathcal{V}_{h+1}(R, B, \lambda), \|\cdot\|_\infty) \cdot \delta/2$ such that

$$
\begin{aligned}
\sup_j \left\|\Sigma_h^{-1/2} \sum_{n=1}^N \mu(\widehat{\omega}_h^n, a_h^n) \Delta^n(V_{h+1,j})\right\|_{\mathcal{H}}^2 \\
\leq H^2 \left(2 \log \frac{2\mathcal{N}(\epsilon, \mathcal{V}_{h+1}(R, B, \lambda), \|\cdot\|_\infty)}{\delta} + \log \det(\lambda I + K_h)\right) \tag{19}
\end{aligned}
$$

holds with probability $1 - \delta/2$. The remaining step is to bound the covering number $\mathcal{N}(\epsilon, \mathcal{V}_{h+1}(R, B, \lambda), \|\cdot\|_\infty)$.

**Lemma 6.** Suppose Assumption 1 and 2 hold. Recall the definition of $\mathcal{V}_h$ in (17). For any $h = 1, \ldots, H$, it holds

$$\log \mathcal{N}(\epsilon, \mathcal{V}_h(R, B, \lambda), \|\cdot\|_\infty)$$

$$\leq \begin{cases} \gamma \log(1 + 6R/\epsilon) + \gamma^2 \log(1 + 6B\sqrt{\gamma}/\epsilon), & \text{(Finite Spectrum)} \\ C_3 \left(\log \frac{3R}{\epsilon_2} + C_4\right)^{\frac{1+\gamma}{\gamma}} + C_6 \left(\log \frac{3B}{\epsilon} + \log \log \frac{3B}{\epsilon} + C_7\right)^{\frac{2+\gamma}{\gamma}}, & \text{(Exponential Decay)} \\ C_5 \left(\frac{3R}{\epsilon}\right)^{\frac{1}{\gamma-1}} \left(\log \frac{12R}{\epsilon} + 1\right) + C_8 \left(\frac{3B}{\epsilon}\right)^{\frac{4}{\gamma-1}} \left(\log \frac{3B}{\epsilon} + C_9\right), & \text{(Polynomial Decay)} \end{cases}$$

$$+ \log\left(1 + \frac{3B}{\epsilon\sqrt{\lambda}}\right),$$

where constants $C_i$ depend on $C, C_1, C_2, \lambda,$ and $\gamma$ in *(Exponential Decay)* and *(Polynomial Decay)*, for $i = 3, \ldots, 10$.

The proof is provided in Appendix C.7. Combining (18), (19) and Lemma 6, we obtain

$$|(C)| \leq \left(H\sqrt{2\log \mathcal{N}(\epsilon, \mathcal{V}_{h+1}(R, B, \lambda), \|\cdot\|_\infty) + 2\log 2/\delta + \log \det(\lambda I + K_h)} + 2\epsilon\lambda^{-1/2}N\right)$$

$$\cdot \left\|\Sigma_h^{-1/2}\mu(\omega, a)\right\|_{\mathcal{H}}. \tag{20}$$

with probability $1 - \delta/2$.

**Step 4: Completing the Proof.** We choose proper $\epsilon, R, B, \lambda,$ and verify condition (9) to finish the proof. We set $\lambda = 1$, and determine $R$ first. By parameterizing $f(\mu(\omega, a)) = \langle \mu(\omega, a), w\rangle$ in (5), we can solve the kernel ridge regression and obtain a closed form solution at step $h$ as

$$\widehat{w}_h = \Sigma_h^{-1}\Phi_h^\top[r_h^1 + \widehat{V}_{h+1}(\widehat{\omega}_{h+1}^1), \ldots, r_h^N + \widehat{V}_{h+1}(\widehat{\omega}_{h+1}^N)]^\top.$$

Recall that by the definition of $\mathcal{V}_h$ in (17), $R$ can be chosen as an upper bound on $\|\widehat{w}_h\|_{\mathcal{H}}$, which is provided in the following Lemma.

**Lemma 7.** For any $\lambda \geq 1$, we have

$$\|\widehat{w}_h\|_{\mathcal{H}} \leq H\lambda^{-1/2}\sqrt{\log \det(I + K_h/\lambda)}.$$

The proof is provided in Appendix C.8. As a result, we choose $R = H\lambda^{-1/2}\sqrt{\log \det(I + K_h/\lambda)}$. We choose $B$ according to the spectrum of kernel $K$.

**Case 1**: *(Finite Spectrum).* We set $B = c\gamma H\sqrt{\log(\max\{d, \gamma\}HN/\delta)}$ for some sufficiently large absolute constant $c$, which is exactly the choice of $\beta$. We first simplify the upper bound on $|(C)|$ in (20). By only keeping the dominating terms in the covering number and effective dimension $\log \det(I + K_h/\lambda)$ (see Lemma 10), for small $\epsilon \in (0, 1)$, we have

$$2\log \mathcal{N}(\epsilon, \mathcal{V}_{h+1}(R, B, \lambda), \|\cdot\|_\infty) + 2\log 2/\delta + \log \det(\lambda I + K_h)$$

$$\leq 4\gamma^2 \log\left(1 + \frac{6\max\{R, B\}\sqrt{\gamma}}{\epsilon}\right) + 2\log 2/\delta + C_{\text{eff-FS}} \cdot \max\{d, \gamma\}\log N$$

$$\leq 4\gamma^2 \log\left(\frac{6cH\gamma^{3/2}\sqrt{\log(\max\{d, \gamma\}HN/\delta)}}{\epsilon\delta}\right) + C_{\text{eff-FS}} \cdot \max\{d, \gamma\}\log N,$$

where the last inequality is valid when constant $c$ is large. Setting $\epsilon = \frac{\gamma H\sqrt{\log(\max\{d, \gamma\}HN/\delta)}}{2N}$, we derive

$$H\sqrt{2\log \mathcal{N}(\epsilon, \mathcal{V}_{h+1}(R, B, \lambda), \|\cdot\|_\infty) + 2\log 2/\delta + \log \det(\lambda I + K_h)} + 2\epsilon\lambda^{-1/2}N$$

$$\overset{(i)}{\leq} 2\gamma H\sqrt{\log\left(12cHN\sqrt{\gamma}/\delta\right)} + H\sqrt{C_{\text{eff-FS}} \cdot \max\{d, \gamma\}\log N} + \gamma H\sqrt{\log(\max\{d, \gamma\}HN/\delta)}$$

$$\overset{(ii)}{\leq} \frac{c}{2}\gamma H\sqrt{\log(\max\{d, \gamma\}HN/\delta)},$$

where inequality $(i)$ follows from $\sqrt{a+b} \leq \sqrt{a} + \sqrt{b}$ and inequality $(ii)$ is always valid when $c$ is properly chosen. Combining (12), (15), and (20), we have

$$
\left| (\mathbb{B}_h \widehat{V}_{h+1})(\omega, a) - (\widehat{\mathbb{B}}_h \widehat{V}_{h+1})(\omega, a) \right|
$$
$$
\leq |(A)| + |(B)| + |(C)|
$$
$$
\leq \left( \left( \sqrt{\frac{1}{m}} + \sqrt{\frac{\log(2/\delta)}{m}} + H \right) \sqrt{\max\{d, \gamma\} \log N} + \frac{c}{2} \gamma H \sqrt{\log(\max\{d, \gamma\} HN/\delta)} \right)
$$
$$
\cdot \left\| \Sigma_h^{-1/2} \mu(\omega, a) \right\|_{\mathcal{H}}
$$
$$
\overset{(i)}{\leq} c\gamma H \sqrt{\log(\max\{d, \gamma\} HN/\delta)} \left\| \Sigma_h^{-1/2} \mu(\omega, a) \right\|_{\mathcal{H}}
$$
$$
= \Gamma_h(\omega, a),
$$

where inequality $(i)$ holds, since $m \geq \log(2/\delta)$ and $c$ is sufficiently large. As a consequence, condition (9) holds true, and we bound $\texttt{SubOpt}(\widehat{\pi}; \omega)$ by

$$
\texttt{SubOpt}(\widehat{\pi}; \omega) \leq 2 \sum_{h=1}^{H} \mathbb{E}_{\pi^*} \left[ \Gamma_h(\omega_h, a_h) \mid \omega_1 = \omega \right].
$$

**Case 2**: *(Exponential Decay)*. We set $B = cH\sqrt{d\left(\log(HN/\delta)\right)^{1+2/\gamma}}$ for a sufficiently large constant $c$. Utilizing similar analysis in **Case 1**, for small $\epsilon \in (0, 1)$ and sufficiently large $c$, we simplify the bound for $|(C)|$:

$$
2 \log \mathcal{N}(\epsilon, \mathcal{V}_{h+1}(R, B, \lambda), \|\cdot\|_\infty) + 2 \log 2/\delta + \log \det(\lambda I + K_h)
$$
$$
\leq (C_3 + C_6) \log \left( \frac{6 \max\{R, B\}}{\epsilon} + (C_4 + C_7) \right)^{\frac{2+\gamma}{\gamma}} + 2 \log 2/\delta + C_{\text{eff-ED}} \cdot d (\log N)^{\frac{1+\gamma}{\gamma}}
$$
$$
\leq (C_3 + C_6) d \log \left( \frac{12 c H \sqrt{d(\log(HN/\delta))^{1+2/\gamma}}}{\epsilon \delta} \right)^{\frac{2+\gamma}{\gamma}} + C_{\text{eff-ED}} \cdot d (\log N)^{\frac{1+\gamma}{\gamma}}.
$$

Choosing $\epsilon = \frac{H\sqrt{d(\log(HN/\delta)^{1+2/\gamma}}}{2N}$, we derive

$$
H\sqrt{2 \log \mathcal{N}(\epsilon, \mathcal{V}_{h+1}(R, B, \lambda), \|\cdot\|_\infty) + 2\log 2/\delta + \log \det(\lambda I + K_h)} + 2\epsilon \lambda^{-1/2} N
$$
$$
\leq \sqrt{2(C_3 + C_6)} H \sqrt{d \log(24cN/\delta)} + H\sqrt{C_{\text{eff-ED}} \cdot d(\log N)^{1+1/\gamma}} + H\sqrt{d(\log(HN/\delta))^{1+2/\gamma}}
$$
$$
\leq \frac{c}{2} H \sqrt{d(\log(HN/\delta))^{1+2/\gamma}}.
$$

Lastly, combining (12), (15), and (20), we have

$$
\left| (\mathbb{B}_h \widehat{V}_{h+1})(\omega, a) - (\widehat{\mathbb{B}}_h \widehat{V}_{h+1})(\omega, a) \right|
$$
$$
\leq |(A)| + |(B)| + |(C)|
$$
$$
\leq \left( \left( \sqrt{\frac{1}{m}} + \sqrt{\frac{\log(2/\delta)}{m}} + H \right) \sqrt{C_{\text{eff-ED}} \cdot d(\log N)^{1+1/\gamma}} + \frac{c}{2} H \sqrt{d(\log(HN/\delta))^{1+2/\gamma}} \right)
$$
$$
\cdot \left\| \Sigma_h^{-1/2} \mu(\omega, a) \right\|_{\mathcal{H}}
$$
$$
\leq cH \sqrt{d(\log(HN/\delta))^{1+2/\gamma}} \left\| \Sigma_h^{-1/2} \mu(\omega, a) \right\|_{\mathcal{H}}
$$
$$
= \Gamma_h(\omega, a).
$$

As a consequence, condition (9) holds true.

**Case 3**: *(Polynomial Decay).* We set $B = cN^{\frac{d+1}{d+\gamma}} H\sqrt{d\log(HN/\delta)}$ for a sufficiently large constant $c$. For small $\epsilon \in (0,1)$ and sufficiently large $c$, we begin with simplifying the bound for $|(C)|$:

$$2\log\mathcal{N}(\epsilon, \mathcal{V}_{h+1}(R, B, \lambda), \|\cdot\|_\infty) + 2\log 2/\delta + \log\det(\lambda I + K_h)$$

$$\leq (C_5 + C_8)\left(\frac{3\max\{R, B\}}{\epsilon}\right)^{4/(\gamma-1)}\log\left(\frac{12\max\{R, B\}}{\epsilon} + C_9 + 1\right) + 2\log 2/\delta$$

$$+ C_{\text{eff-PD}} \cdot N^{\frac{d+1}{d+\gamma}} d\log N$$

$$\leq (C_5 + C_8)\left(\frac{6cN^{\frac{d+1}{d+\gamma}} H\sqrt{d\log(HN/\delta)}}{\epsilon}\right)^{\frac{4}{\gamma-1}} + C_{\text{eff-PD}} \cdot N^{\frac{d+1}{d+\gamma}} d\log N.$$

Choosing $\epsilon = \frac{1}{2}N^{-1+\frac{d+1}{d+\gamma}} H\sqrt{d\log(HN/\delta)}$, we derive

$$H\sqrt{2\log\mathcal{N}(\epsilon, \mathcal{V}_{h+1}(R, B, \lambda), \|\cdot\|_\infty) + 2\log 2/\delta + \log\det(\lambda I + K_h)} + 2\epsilon\lambda^{-1/2}N$$

$$\leq \sqrt{2(C_5 + C_8)} H(12cN)^{\frac{2}{\gamma-1}} + H\sqrt{C_{\text{eff-PD}} \cdot N^{\frac{d+1}{d+\gamma}} d\log N} + N^{\frac{d+1}{d+\gamma}} H\sqrt{d\log(HN/\delta)}$$

$$\overset{(i)}{\leq} \frac{c}{2} HN^{\frac{d+1}{d+\gamma}} H\sqrt{d\log(HN/\delta)},$$

where in $(i)$, we have $\frac{2}{\gamma-1} \leq \frac{d+1}{d+\gamma}$ for $\gamma \geq 3 + 4/(d-1)$. Lastly, combining (12), (15), and (20), we have

$$\left|(\mathbb{B}_h\widehat{V}_{h+1})(\omega, a) - (\widehat{\mathbb{B}}_h\widehat{V}_{h+1})(\omega, a)\right|$$

$$\leq |(A)| + |(B)| + |(C)|$$

$$\leq \left(\left(\sqrt{\frac{1}{m}} + \sqrt{\frac{\log(2/\delta)}{m}} + H\right)\sqrt{C_{\text{eff-PD}} \cdot N^{\frac{d+1}{d+\gamma}} d\log N} + \frac{c}{2} HN^{\frac{d+1}{d+\gamma}} H\sqrt{d\log(HN/\delta)}\right)$$

$$\cdot \left\|\Sigma_h^{-1/2}\mu(\omega, a)\right\|_{\mathcal{H}}$$

$$\leq cHN^{\frac{d+1}{d+\gamma}} H\sqrt{d\log(HN/\delta)} \left\|\Sigma_h^{-1/2}\mu(\omega, a)\right\|_{\mathcal{H}}$$

$$= \Gamma_h(\omega, a).$$

As a consequence, condition (9) holds true. Therefore, we complete the proof. $\qquad\square$

## B   Proof of Corollary 1

*Proof.* Given Theorem 1, we only need to show the convergence of $\Gamma_h$ under Assumption 4. In particular, we show that $\|\Sigma_h^{-1/2}\mu(\omega, a)\|_{\mathcal{H}}$ converges at a rate of $1/\sqrt{N}$. We denote $A_n = \mu(\omega_h^n, a_h^n) \otimes \mu(\omega_h^n, a_h^n) - \mathbb{E}_{\bar\pi}[\mu(\omega_h, a_h) \otimes \mu(\omega_h, a_h)]$, which verifies $\mathbb{E}_{\bar\pi}[A_n] = 0$. We denote $Z = \sum_{n=1}^N A_n$, and distinguish three cases according to the spectrum of $K$. (In the following proof, we denote $\|\cdot\|_{\text{op}}$ as the operator norm.)

**Case 1**: *(Finite Spectrum).* Using the same argument in Lemma 9, we represent $\mu(\omega_h^n, a) \otimes \mu(\omega_h^n, a)$ as a $\gamma \times \gamma$ matrix $W_h^n$, since kernel $K$ has a $\gamma$-finite spectrum. We also denote $W_h = \mathbb{E}_{\bar\pi}[\mu(\omega_h, a_h) \otimes \mu(\omega_h, a_h)]$. We keep the notation $A_n$ and $Z$ unchanged, yet overload the two with $A_n = W_h^n - W_h$ and $Z = \sum_{n=1}^N A_n$.

By the boundedness of kernel, we deduce that the matrix operator norm $\|W_h\|_{\text{op}}$ is bounded by 1. Meanwhile, the operator norm of $A_n$ is bounded by $\|A_n\|_{\text{op}} \leq \|W_h^n\|_{\text{op}} + \|W_h\|_{\text{op}} \leq 2$, using the triangle inequality. Furthermore, we have

$$\left\|\mathbb{E}_{\bar\pi}[ZZ^\top]\right\|_{\text{op}} = N\left\|A_n A_n^\top\right\|_{\text{op}} \leq N\|A_n\|_{\text{op}}\left\|A_n^\top\right\|_{\text{op}} \leq 4N,$$

and similarly

$$\left\|\mathbb{E}_{\bar\pi}[Z^\top Z]\right\|_{\text{op}} = N\left\|A_n^\top A_n\right\|_{\text{op}} \leq 4N.$$

Applying matrix Bernstein inequality, for any $t > 0$, we have

$$\mathbb{P}\left(\|Z\|_{\mathrm{op}} \geq t\right) \leq 2\gamma \exp\left(-\frac{t^2/2}{4N + 2t/3}\right).$$

Taking $t = \sqrt{10N \log(4\gamma H/\delta)}$, for sufficiently large $N \geq 5\log(4\gamma H/\delta)$, we obtain

$$\|Z\|_{\mathrm{op}} \leq \sqrt{10N \log(4\gamma H/\delta)} \quad \text{holds with probability} \quad 1 - \delta/2H.$$

**Case 2**: *(Exponential Decay).* We truncate the spectrum of kernel $K$ at $M$, where the positive integer $M$ will be determined later. Denote $\bar{\mu}$ as the truncated version of $\mu$. Using the truncation error in Lemma 9 with $\lambda = 1$, we bound

$$e_M \overset{\triangle}{=} \|\mu(\omega, a) \otimes \mu(\omega, a) - \bar{\mu}(\omega, a) \otimes \bar{\mu}(\omega, a)\|_{\mathrm{op}}$$

$$\leq \begin{cases} 2\sqrt{\frac{C_1}{C_2}\exp(-C_2 M^\gamma)} & \gamma \geq 1 \\ 2\sqrt{\frac{C_1 M^{1-\gamma}}{C_2\gamma}\exp(-C_2 M^\gamma)} & \gamma \in (0,1) \end{cases}.$$

We let $\bar{A}_n = \bar{\mu}(\omega_h^n, a_h^n) \otimes \bar{\mu}(\omega_h^n, a_h^n) - \mathbb{E}_{\bar{\pi}}[\bar{\mu}(\omega_h, a_h) \otimes \bar{\mu}(\omega_h, a_h)]$ and $\bar{Z} = \sum_{n=1}^N \bar{A}_n$. Then we can derive

$$\mathbb{P}\left(\|Z\|_{\mathrm{op}} \geq t\right) \leq \mathbb{P}\left(\|\bar{Z}\|_{\mathrm{op}} \geq t - 2Ne_M\right)$$

$$\leq 2M\exp\left(-\frac{(t - 2Ne_M)^2/2}{4N + 2(t - 2Ne_M)/3}\right).$$

We choose $M = \frac{c_0}{C_2}(\log\frac{C_1 N}{C_2})^{1/\gamma}$ for some absolute constant $c_0 \leq 2$ and $t = 8\sqrt{N\log(4Hd_{\mathrm{eff}}/\delta)}$, which gives rise to

$$t - 2Ne_M = 8\sqrt{N\log(4Hd_{\mathrm{eff}}/\delta)} - 4\sqrt{N} \leq \sqrt{10N\log(4Hd_{\mathrm{eff}}/\delta)}.$$

For sufficiently large $N \geq 5\log(4Hd_{\mathrm{eff}}/\delta)$, we obtain

$$\|Z\|_{\mathrm{op}} \leq 8\sqrt{N\log(4Hd_{\mathrm{eff}}/\delta)} \quad \text{holds with probability} \quad 1 - \delta/2H.$$

**Case 3**: *(Polynomial Decay).* We consider truncating the spectrum at $M$ again. Using the truncation error of polynomial decay case in Lemma 9, we have

$$\mathbb{P}\left(\|Z\|_{\mathrm{op}} \geq t\right) \leq \mathbb{P}\left(\|\bar{Z}\|_{\mathrm{op}} \geq t - 4N\sqrt{(\gamma-1)^{-1}CM^{-\gamma+1}}\right)$$

$$\leq 2M\exp\left(-\frac{\left(t - 4N\sqrt{(\gamma-1)^{-1}CM^{-\gamma+1}}\right)^2/2}{4N + 2\left(t - 4N\sqrt{(\gamma-1)^{-1}CM^{-\gamma+1}}\right)/3}\right).$$

We choose $M = \left(\frac{CN}{\gamma-1}\right)^{\frac{1}{\gamma-1}}$ and $t = 8\sqrt{N\log(4Hd_{\mathrm{eff}}/\delta)}$, which gives rise to

$$t - 4N\sqrt{(\gamma-1)^{-1}CM^{-\gamma+1}} = 8\sqrt{N\log(4Hd_{\mathrm{eff}}/\delta)} - 4\sqrt{N} \leq \sqrt{10N\log(4Hd_{\mathrm{eff}}/\delta)}.$$

Further, when $\gamma \geq 2 + 1/d$, we have $\frac{d+1}{d+\gamma} > \frac{1}{\gamma-1}$, which implies $M \leq d_{\mathrm{eff}}$. Thus, for $N \geq 5\log(4Hd_{\mathrm{eff}}/\delta)$, we have

$$\|Z\|_{\mathrm{op}} \leq 8\sqrt{N\log(4Hd_{\mathrm{eff}}/\delta)} \quad \text{holds with probability} \quad 1 - \delta/2H.$$

We rewrite $Z$ using the covariance operator $\Sigma_h$ as

$$Z = \Sigma_h - \lambda I_{\mathcal{H}_K} - N\mathbb{E}_{\bar{\pi}}[\mu(\omega_h, a_h) \otimes \mu(\omega_h, a_h)]$$

$$+ \underbrace{\sum_{n=1}^N \{\mu(\widehat{\omega}_h^n, a_h^n) \otimes \mu(\widehat{\omega}_h^n, a_h^n) - \mu(\omega_h^n, a_h^n) \otimes \mu(\omega_h^n, a_h^n)\}}_{\mathcal{E}}.$$

We bound the operator norm of $\mathcal{E}$ by

$$
\begin{aligned}
\|\mathcal{E}\|_{\mathrm{op}} &= \sup_{\|f\|_{\mathcal{H}} \leq 1} \|\mathcal{E}f\|_{\mathcal{H}} \\
&\leq N \|\mu(\widehat{\omega}_h^n, a_h^n) \langle \mu(\widehat{\omega}_h^n, a_h^n), f \rangle - \mu(\omega_h^n, a_h^n) \langle \mu(\omega_h^n, a_h^n), f \rangle \|_{\mathcal{H}} \\
&\leq 2N \|\mu(\widehat{\omega}_h^n, a_h^n) - \mu(\omega_h, a_h)\|_{\mathcal{H}}.
\end{aligned}
$$

By Lemma 3, we have $\|\mu(\widehat{\omega}_h^n, a_h^n) - \mu(\omega_h, a_h)\|_{\mathcal{H}} \leq \sqrt{\frac{2}{m}} + \sqrt{\frac{2\log(4H/\delta)}{m}}$ with probability $1 - \delta/4H$. Thus, for sufficiently large $m \geq 32N^2 \log(4H/\delta)$, $\|\mathcal{E}\|_{\mathrm{op}} \leq \sqrt{N}$ with probability $1 - \delta/4H$. Therefore, with probability $1 - \frac{3}{4}\delta$, we deduce that

$$
\begin{aligned}
\left\| \frac{1}{N}(\Sigma_h - \lambda I) - \mathbb{E}_{\bar{\pi}}[\mu(\omega_h, a_h) \otimes \mu(\omega_h, a_h)] \right\|_{\mathrm{op}} &\leq \frac{1}{N} \|Z\|_{\mathrm{op}} + \frac{1}{N} \|\mathcal{E}\|_{\mathrm{op}} \\
&\leq 8\sqrt{\frac{\log(4d_{\mathrm{eff}}H/\delta)}{N}} + \sqrt{\frac{1}{N}}
\end{aligned}
$$

holds simultaneously for all $h = 1, \ldots, H$. For sufficiently large $N \geq \frac{1024}{c_{\min}^2} \log(4d_{\mathrm{eff}}H/\delta)$, we have

$$
\frac{1}{N}\Sigma_h \succeq \mathbb{E}_{\bar{\pi}}[\mu(\omega_h, a_h) \otimes \mu(\omega_h, a_h)] - \left( 8\sqrt{\frac{\log(4d_{\mathrm{eff}}H/\delta)}{N}} + \sqrt{\frac{1}{N}} \right) \cdot I_{\mathcal{H}_K} \succeq \frac{c_{\min}}{2} \cdot I_{\mathcal{H}_K}.
$$

This further implies

$$
\left\| \Sigma_h^{-1/2} \mu(\omega, a) \right\|_{\mathcal{H}} \leq \|\mu(\omega, a)\|_{\mathcal{H}} \left\| \Sigma_h^{-1} \right\|_{\mathrm{op}} \leq \frac{2}{c_{\min}\sqrt{N}}.
$$

Combining with Theorem 1 and taking $\delta = \delta/4$ therein, we have

$$
\begin{aligned}
\mathtt{SubOpt}(\widehat{\pi}; \omega) &\leq 2 \sum_{h=1}^{H} [\Gamma_h(\omega_h, a_h) \mid \omega_1 = \omega] \\
&\leq 2\beta \sum_{h=1}^{H} \left[ \frac{2}{c_{\min}} N^{-1/2} \mid \omega_1 = \omega \right] \\
&= \mathcal{O}\left( H^2 d_{\mathrm{eff}} \sqrt{\frac{\log(d_{\mathrm{eff}}HN/\delta)}{N}} \right),
\end{aligned}
$$

where in the last inequality, we substitute into the choice of $\beta$ and note that by Lemma 10, $\beta = \mathcal{O}(Hd_{\mathrm{eff}}\sqrt{\log(d_{\mathrm{eff}}HN/\delta)})$. The proof is complete. $\qquad\square$

## C  Proofs of Supporting Lemmas for Theorem 1

We provide proofs of technical lemmas for establishing our main results.

### C.1  Proof of Proposition 1

*Proof.* We first show $\mathbb{B}_h g$ and $\mathbb{P}_h g$ can be parameterized in $\mathcal{H}_K$. Using the definition in (2), we derive

$$
\begin{aligned}
(\mathbb{P}_h g)(\omega, a) &= \mathbb{E}\left[ g(\omega_{h+1}) \mid \omega_h = \omega, a_h = a \right] \\
&= \int_{\Omega} g(x) p_h(x \mid \omega, a) dx \\
&= \int_{\Omega} g(x) \langle \mu(\omega, a), v_h(x) \rangle dx \\
&= \left\langle \mu(\omega, a), \int_{\Omega} g(x) v_h(x) dx \right\rangle \in \mathcal{H}_K.
\end{aligned}
$$

Similarly, for $\mathbb{B}_h g$, we have

$$
\begin{aligned}
(\mathbb{B}_h g)(\omega, a) &= (\mathbb{P}_h g)(\omega, a) + r_h(\omega, a) \\
&= \left\langle \mu(\omega, a), \int_\Omega g(x) v_h(x) dx \right\rangle + \langle \mu(\omega, a), \theta_h \rangle \\
&= \left\langle \mu(\omega, a), \int_\Omega g(x) v_h(x) dx + \theta_h \right\rangle \in \mathcal{H}_K.
\end{aligned}
$$

The proof is complete. $\qquad \square$

## C.2 Proof of Lemma 1

*Proof.* We write $\mathtt{SubOpt}(\widehat{\pi}; \omega)$ as

$$
\mathtt{SubOpt}(\pi; \omega) = V_1^{\pi^*}(\omega) - V_1(\omega) + V_1(\omega) - V_1^\pi(\omega), \tag{21}
$$

where $V_h = \langle Q_h, \pi \rangle_{\mathcal{A}}$. Note the difference between $V_1$ and $V_1^\pi$, where in the former, each $V_h$ is computed from a given $Q$-function $Q_h$. By the extended value difference in Section B.1 of Cai et al. (2020) (see also Lemma A.1 in Jin et al. (2020b)), we have

$$
\begin{aligned}
V_1(\omega) - V_1^\pi(\omega) &= \mathbb{E}_\pi \left[ \langle Q_1(\omega_1, \cdot), \pi(\cdot \mid \omega_1) \rangle_{\mathcal{A}} - \langle Q_1^\pi(\omega_1, \cdot), \pi(\cdot \mid \omega_1) \rangle_{\mathcal{A}} \mid \omega_1 = \omega \right] \\
&= \mathbb{E}_\pi \Big[ \langle Q_1(\omega_1, \cdot), \pi(\cdot \mid \omega_1) \rangle_{\mathcal{A}} - \langle (\mathbb{B}_1 V_2)(\omega_1, \cdot), \pi(\cdot \mid \omega_1) \rangle_{\mathcal{A}} \\
&\quad + \langle (\mathbb{B}_1 V_2)(\omega_1, \cdot), \pi(\cdot \mid \omega_1) \rangle_{\mathcal{A}} - \langle Q_1^\pi(\omega, \cdot), \pi(\cdot \mid \omega) \rangle_{\mathcal{A}} \mid \omega_1 = \omega \Big] \\
&= \mathbb{E}_\pi \left[ Q_1(\omega_1, a_1) - (\mathbb{B}_1 V_2)(\omega_1, a_1) \mid \omega_1 = \omega \right] + \mathbb{E}_\pi \left[ V_2(\omega_2) - V_2^\pi(\omega_2) \mid \omega_1 = \omega \right] \\
&= \cdots \\
&= \sum_{h=1}^H \mathbb{E}_\pi \left[ Q_h(\omega_h, a_h) - (\mathbb{B}_h V_{h+1})(\omega_h, a_h) \mid \omega_1 = \omega \right]. \tag{22}
\end{aligned}
$$

Analogously, we derive

$$
\begin{aligned}
V_1^{\pi^*}(\omega) - V_1(\omega) &= \sum_{h=1}^H \mathbb{E}_{\pi^*} \left[ \langle Q_h(\omega_h, \cdot), \pi_h(\cdot \mid \omega_h) - \pi^*(\cdot \mid \omega_h) \rangle \mid \omega_1 = \omega \right] \\
&\quad + \sum_{h=1}^H \mathbb{E}_{\pi^*} \left[ Q_h(\omega_h, a_h) - (\mathbb{B}_h V_{h+1})(\omega_h, a_h) \mid \omega_1 = \omega \right]. \tag{23}
\end{aligned}
$$

Substituting (22) and (23) into (21), we obtain the desired decomposition in Lemma 1. $\qquad \square$

## C.3 Proof of Lemma 2

*Proof.* We first prove the left inequality, i.e.,

$$
0 \leq (\mathbb{B}_h \widehat{V}_{h+1})(\omega, a) - \widehat{Q}_h(\omega, a).
$$

Conditioned on the event $\left| (\mathbb{B}_h \widehat{V}_{h+1})(\omega, a) - (\widehat{\mathbb{B}}_h \widehat{V}_{h+1})(\omega, a) \right| \leq \Gamma_h(\omega, a)$, we have

$$
(\mathbb{B}_h \widehat{V}_{h+1})(\omega, a) - \widehat{Q}_h(\omega, a) \overset{(i)}{\geq} (\mathbb{B}_h \widehat{V}_{h+1})(\omega, a) - \widetilde{Q}_h(\omega, a) + \Gamma(\omega, a) \geq 0,
$$

where inequality $(i)$ follows from $\widehat{Q}$ is a bounded truncated version of $\widetilde{Q}_h$. Therefore, the left inequality holds for any $(\omega, a)$. Next, we show the right inequality, i.e.,

$$
(\mathbb{B}_h \widehat{V}_{h+1})(\omega, a) - \widehat{Q}_h(\omega, a) \leq 2\Gamma_h(\omega, a).
$$

Observe $\widehat{V}_{h+1} \leq H - h$ in Algorithm 1. Combining with $|r_h| \leq 1$, we have

$$
(\mathbb{B}_h \widehat{V}_{h+1})(\omega, a) \leq H - h + 1.
$$

This implies $(\widehat{\mathbb{B}}_h \widehat{V}_{h+1}) \le H - h + 1 - \Gamma_h$. Therefore, we deduce

$$\widehat{Q}_h \ge \widetilde{Q}_h - \Gamma_h = (\widehat{\mathbb{B}}_h \widehat{V}_{h+1}) - \Gamma_h.$$

Since we have proved $(\mathbb{B}_h \widehat{V}_{h+1})(\omega, a) - \widehat{Q}_h(\omega, a) \ge 0$, we derive

$$(\mathbb{B}_h \widehat{V}_{h+1})(\omega, a) - \widehat{Q}_h(\omega, a) \le (\mathbb{B}_h \widehat{V}_{h+1})(\omega, a) - (\widehat{\mathbb{B}}_h \widehat{V}_{h+1})(\omega, a) + \Gamma_h(\omega, a) \le 2\Gamma_h(\omega, a).$$

The proof is complete. $\qquad\square$

## C.4    Proof of Lemma 3

*Proof.* We denote

$$g(\widehat{\omega}_m, a) = \|\mu(\widehat{\omega}_m, a) - \mu(\omega, a)\|_{\mathcal{H}}.$$

Consider two meta states $\widehat{\omega}_m = s_0 \times \widehat{d}_s$ and $\widehat{\omega}' = s_0 \times \widehat{d}'_s$ with only the $k$-th agent ($k \ge 1$) having distinct states $s_k$ and $(s_k)'$, respectively. We bound the difference in function value:

$$\begin{aligned}
|g(\widehat{\omega}_m, a) - g(\widehat{\omega}'_m, a)| &= \|\mu(\widehat{\omega}_m, a) - \mu(\omega, a)\|_{\mathcal{H}_K} - \|\mu(\widehat{\omega}'_m, a) - \mu(\omega, a)\|_{\mathcal{H}} \\
&\le \|\mu(\widehat{\omega}_m, a) - \mu(\widehat{\omega}'_m, a)\|_{\mathcal{H}} \\
&\le \frac{1}{m} \|\psi(s_0, s_k, a) - \psi(s_0, s'_k, a)\|_{\mathcal{H}} \\
&\le \frac{2}{m}.
\end{aligned}$$

By Mcdiarmid's inequality, for any $\delta_0 > 0$, we have

$$g(\widehat{\omega}_m, a) \le \mathbb{E}[g(\widehat{\omega}_m, a)] + \delta_0 \quad \text{with probability at least} \quad 1 - \exp\left(-\frac{\delta_0^2 m}{2}\right). \qquad (24)$$

It remains to bound $\mathbb{E}[g(\widehat{\omega}_m, a)]$. Some algebraic manipulation gives rise to

$$\begin{aligned}
\mathbb{E}[g(\widehat{\omega}_m, a)] &\overset{(i)}{\le} \sqrt{\mathbb{E}\left[\|\mu(\widehat{\omega}_m, a) - \mu(\omega, a)\|_{\mathcal{H}}^2\right]} \\
&= \sqrt{\mathbb{E}\left[\frac{1}{m}\sum_{i=1}^{m} \psi(s_0, s_i, a) - \mu(\omega, a)\right]} \\
&= \sqrt{\frac{1}{m}\mathbb{E}_{s \sim d_s, s' \sim d'_s}[k((s_0, s, a), (s_0, s, a)) - k((s_0, s, a), (s_0, s', a))]} \\
&\le \sqrt{2/m},
\end{aligned} \qquad (25)$$

where inequality $(i)$ follows from Jensen's inequality. (A similar computation appears in Theorem 15 of Altun and Smola (2006).) Substituting (25) into the right-hand side of (24), with probability at least $1 - \exp\left(-\frac{\delta_0^2 m}{2}\right)$, we have

$$\|\mu(\widehat{\omega}_m, a) - \mu(\omega, a)\|_{\mathcal{H}} \le \sqrt{2/m} + \delta_0.$$

Taking $\delta_0 = \sqrt{\frac{2\log(1/\delta_A)}{m}}$, we deduce

$$\|\mu(\widehat{\omega}_m, a) - \mu(\omega, a)\|_{\mathcal{H}} \le \sqrt{\frac{2}{m}} + \sqrt{\frac{2\log(1/\delta_A)}{m}} \quad \text{with probability at least} \quad 1 - \delta_A.$$

The proof is complete. $\qquad\square$

## C.5  Proof of Lemma 4

*Proof.* Using identity (16), we bound $(A)$ as

$$
\begin{aligned}
|(A)| &= \left| \phi_h(\omega, a)^\top \Lambda_h^{-1} \left[ \widehat{r}_h^1 - r_h^1, \dots, \widehat{r}_h^N - r_h^N \right]^\top \right| \\
&= \left| \left\langle \mu(\omega, a), \Sigma_h^{-1} \Phi_h^\top \left[ \widehat{r}_h^1 - r_h^1, \dots, \widehat{r}_h^N - r_h^N \right]^\top \right\rangle \right| \\
&\overset{(i)}{\leq} \|\theta_h\|_{\mathcal{H}} \left( \sqrt{\frac{2}{m}} + \sqrt{\frac{2\log(1/\delta_A)}{m}} \right) \sum_{n=1}^N \left| \left\langle \mu(\omega, a), \Sigma_h^{-1} \mu(\widehat{\omega}_h^n, a_h^n) \right\rangle \right| \\
&\overset{(ii)}{\leq} \left( \sqrt{\frac{2}{m}} + \sqrt{\frac{2\log(1/\delta_A)}{m}} \right) \left\| \Sigma_h^{-1/2} \mu(\omega, a) \right\|_{\mathcal{H}} \sqrt{\sum_{n=1}^N \left\langle \mu(\widehat{\omega}_h^n, a_h^n), \Sigma_h^{-1} \mu(\widehat{\omega}_h^n, a_h^n) \right\rangle}, \quad (26)
\end{aligned}
$$

where inequality $(i)$ invokes (11) and holds with probability $1 - \delta_A$, inequality $(ii)$ follows from Cauchy-Schwarz inequality. It remains to bound $\sum_{n=1}^N \left\langle \psi_h(\mu_{\widehat{\zeta}_h^n}), \Sigma_h^{-1} \psi_h(\mu_{\widehat{\zeta}_h^n}) \right\rangle_{\mathcal{H}}$. By Lemma 11 in Abbasi-Yadkori et al. (2011) (see also Lemma E.3 in Yang et al. (2020b)), we have

$$
\sum_{n=1}^N \left\langle \mu(\widehat{\omega}_h^n, a_h^n), \Sigma_h^{-1} \mu(\widehat{\omega}_h^n, a_h^n) \right\rangle_{\mathcal{H}} \leq 2 \log \det \left( I + K_h/\lambda \right). \quad (27)
$$

Substituting (27) into (26), we obtain

$$
|(A)| \leq 2 \left( \sqrt{\frac{1}{m}} + \sqrt{\frac{\log(1/\delta_A)}{m}} \right) \sqrt{\log \det(I + K_h/\lambda)} \left\| \Sigma_h^{-1/2} \mu(\omega, a) \right\|_{\mathcal{H}}
$$

with probability $1 - \delta_A$. $\qquad \square$

## C.6  Proof of Lemma 5

*Proof.* The proof is based on concentration of measure in self-normalizing sequences. Let $\mathcal{F}_{h,\tau}$ be the $\sigma$-algebra generated by data $\{(s_h^n, a_h^n, r_h^n)\}_{n=1}^\tau$. By the Markov property and definition of Bellman operator, we have $\mathbb{E}[\Delta^\tau(V_{h+1}) \mid \mathcal{F}_{h,\tau-1}] = 0$. Moreover, $\Delta^n(V_{h+1}) \leq H$, since the reward function is bounded by 1 in Assumption 3. By Theorem 1 in Chowdhury and Gopalan (2017), with probability at least $1 - \delta$, for any $\eta > 0$, we have

$$
\Delta(V_{h+1})^\top ((K_h + \eta I)^{-1} + I)^{-1} \Delta(V_{h+1}) \leq 2H^2 \log \det((1 + \eta)I + K_h) + 2H^2 \log \frac{1}{\delta}, \quad (28)
$$

where $\Delta(V_{h+1}) = [\Delta^1(V_{h+1}), \dots, \Delta^N(V_{h+1})]^\top$. In the remaining of the proof, we take $\eta = \lambda - 1$ and show

$$
\left\| \Sigma_h^{-1/2} \sum_{n=1}^N \mu(\widehat{\omega}_h^n, a_h^n) \Delta^n(V_{h+1}) \right\|_{\mathcal{H}}^2 \leq \Delta(V_{h+1})^\top ((K_h + (\lambda-1)I)^{-1} + I)^{-1} \Delta(V_{h+1}).
$$

As a consequence, (28) is an upper bound of $\left\| \Sigma_h^{-1/2} \sum_{n=1}^N \mu(\widehat{\omega}_h^n, a_h^n) \Delta^n(V_{h+1}) \right\|_{\mathcal{H}}^2$, which is the desired result.

Using vector notations, we rewrite $\Sigma_h^{-1/2} \sum_{n=1}^N \mu(\widehat{\omega}_h^n, a_h^n) \Delta^n(V_{h+1})$ as $\Sigma_h^{-1/2} \Phi_h^\top \Delta(V_{h+1})$. Then we derive

$$
\begin{aligned}
\left\| \Sigma_h^{-1/2} \sum_{n=1}^N \mu(\widehat{\omega}_h^n, a_h^n) \Delta^n(V_{h+1}) \right\|_{\mathcal{H}}^2 &= \left\| \Sigma_h^{-1/2} \Phi_h^\top \Delta(V_{h+1}) \right\|_{\mathcal{H}}^2 \\
&= \Delta(V_{h+1})^\top \left\langle \Phi_h, \Sigma_h^{-1} \Phi_h^\top \right\rangle \Delta(V_{h+1}) \\
&\overset{(i)}{=} \Delta(V_{h+1})^\top \left\langle \Phi_h, \Phi_h^\top \right\rangle (\lambda I + K_h)^{-1} \Delta(V_{h+1}) \\
&= \Delta(V_{h+1})^\top K_h (\lambda I + K_h)^{-1} \Delta(V_{h+1}),
\end{aligned}
$$

where equality $(i)$ invokes the identity in (13). Now we need to show $\Delta(V_{h+1})^\top((K_h + (\lambda - 1)I)^{-1} + I)^{-1}\Delta(V_{h+1}) \geq \Delta(V_{h+1})^\top K_h(\lambda I + K_h)^{-1}\Delta(V_{h+1})$. Indeed, by the matrix inversion lemma, we have

$$
\begin{aligned}
\Delta(V_{h+1})^\top((K_h + (\lambda - 1)I)^{-1} + I)^{-1}\Delta(V_{h+1}) &= \Delta(V_{h+1})^\top(I - (K_h + \lambda I)^{-1})\Delta(V_{h+1}) \\
&\geq \Delta(V_{h+1})^\top(I - \lambda(K_h + \lambda I)^{-1})\Delta(V_{h+1}) \\
&= \Delta(V_{h+1})^\top K_h(\lambda I + K_h)^{-1}\Delta(V_{h+1}).
\end{aligned}
$$

The proof is complete. $\qquad\square$

### C.7  Proof of Lemma 6

*Proof.* We reduce the covering of $\mathcal{V}_h(R, B, \lambda)$ to Cartesian product of coverings on $\theta$, $\beta$, and $\Sigma$. Specifically, let $f_1, f_2$ be two elements in $\mathcal{H}_h$. We denote

$$
f_1(\omega, a) = \max_{a \in \mathcal{A}}\left[\min\left\{\langle\mu(\omega, a), \theta_1\rangle - \beta_1\sqrt{\langle\mu(\omega, a), \Sigma_1\mu(\omega, a)\rangle}, H - h + 1\right\}^+\right] \quad \text{and}
$$

$$
f_2(\omega, a) = \max_{a \in \mathcal{A}}\left[\min\left\{\langle\mu(\omega, a), \theta_2\rangle - \beta_2\sqrt{\langle\mu(\omega, a), \Sigma_2\mu(\omega, a)\rangle}, H - h + 1\right\}^+\right].
$$

We evaluate the difference between $f_1$ and $f_2$:

$$
\begin{aligned}
&\|f_1 - f_2\|_\infty \\
&\overset{(i)}{\leq} \sup_{(\omega, a)}\left|\langle\mu(\omega, a), \theta_1 - \theta_2\rangle - \left(\beta_1\sqrt{\langle\mu(\omega, a), \Sigma_1^{-1}\mu(\omega, a)\rangle} - \beta_2\sqrt{\langle\mu(\omega, a), \Sigma_2^{-1}\mu(\omega, a)\rangle}\right)\right| \\
&\leq \sup_{(\omega, a)}|\langle\mu(\omega, a), \theta_1 - \theta_2\rangle| + \sup_{(\omega, a)}|\beta_1 - \beta_2|\sqrt{\langle\mu(\omega, a), \Sigma_1\mu(\omega, a)\rangle} \\
&\quad + \sup_{(\omega, a)}\beta_2\left|\sqrt{\langle\mu(\omega, a), \Sigma_1\mu(\omega, a)\rangle} - \sqrt{\langle\mu(\omega, a), \Sigma_2\mu(\omega, a)\rangle}\right| \\
&\overset{(ii)}{\leq} \|\theta_1 - \theta_2\|_{\mathcal{H}} + \lambda^{-1/2}|\beta_1 - \beta_2| \\
&\quad + B\sup_{(\omega, a)}\left|\sqrt{\langle\mu(\omega, a), \Sigma_1\mu(\omega, a)\rangle} - \sqrt{\langle\mu(\omega, a), \Sigma_2\mu(\omega, a)\rangle}\right| \\
&\leq \|\theta_1 - \theta_2\|_{\mathcal{H}} + \lambda^{-1/2}|\beta_1 - \beta_2| \\
&\quad + B\sup_{(\omega, a)}\sqrt{|\langle\mu(\omega, a), (\Sigma_1 - \Sigma_2)\mu(\omega, a)\rangle|},
\end{aligned}
\tag{29}
$$

where inequality $(i)$ removes the truncation operation in $f_1, f_2$, and inequality $(ii)$ follows from $\sigma_1 \succeq \lambda I_{\mathcal{H}_K}$. Decomposition (29) suggests that an $\epsilon$-covering of $\mathcal{V}_h$ can be constructed from the Cartesian product of an $\epsilon/3$-covering on $\mathcal{F}_1 = \{\theta : \|\theta\|_{\mathcal{H}} \leq R\}$, an $\epsilon\sqrt{\lambda}/3$-covering on $\mathcal{F}_2 = \{\beta : 0 \leq \beta \leq B\}$, and an $\epsilon/(3B)$-covering on $\mathcal{F}_3 = \{\Sigma : \lambda^{-1}I_{\mathcal{H}_K} \succeq \Sigma \succeq 0\}$. Correspondingly, the covering number of $\mathcal{V}_h$ is the product

$$
\mathcal{N}(\epsilon, \mathcal{V}_h(R, B, \lambda), \|\cdot\|_\infty) = \mathcal{N}(\epsilon/3, \mathcal{F}_1, \|\cdot\|_\infty) \cdot \mathcal{N}(\epsilon\sqrt{\lambda}/3, \mathcal{F}_2, \|\cdot\|_\infty) \cdot \mathcal{N}(\epsilon/(3B), \mathcal{F}_3, \|\cdot\|_\infty).
$$

By Lemma 8, we have

$$
\log\mathcal{N}(\epsilon/3, \mathcal{F}_1, \|\cdot\|_\infty) \leq \begin{cases} \gamma\log(1 + 6R/\epsilon), & (Finite\ Spectrum) \\ C_3\left(\log(3R/\epsilon) + C_4\right)^{1+1/\gamma}, & (Exponential\ Decay) \\ C_5(3R/\epsilon)^{2/(\gamma-1)}\log(1 + 12R/\epsilon), & (Polynomial\ Decay) \end{cases}.
$$

A direct discretization yields

$$
\log\mathcal{N}(\epsilon\sqrt{\lambda}/3, \mathcal{F}_2, \|\cdot\|_\infty) \leq \log\left(1 + \frac{3B}{\epsilon\sqrt{\lambda}}\right).
$$

By Lemma 9, we have

$$\log \mathcal{N}(\epsilon/(3B), \mathcal{F}_3, \|\cdot\|_\infty)$$
$$\leq \begin{cases} \gamma^2 \log(1 + 6B\sqrt{\gamma}/(\lambda\epsilon)), & (Finite\ Spectrum) \\ C_6 \left(\log(3B/\epsilon) + \log\log(3B/\epsilon) + C_7\right)^{1+2/\gamma}, & (Exponential\ Decay) \, . \\ C_8(3B/\epsilon)^{4/(\gamma-1)} \left(\log(3B/\epsilon) + C_9\right), & (Polynomial\ Decay) \end{cases}$$

Combining all these covering numbers, we deduce

$$\log \mathcal{N}(\epsilon, \mathcal{V}_h(R, B, \lambda), \|\cdot\|_\infty)$$
$$\leq \begin{cases} \gamma \log(1 + 6R/\epsilon) + \gamma^2 \log(1 + 6B\sqrt{\gamma}/\epsilon), & (Finite\ Spectrum) \\ C_3 \left(\log\frac{3R}{\epsilon_2} + C_4\right)^{\frac{1+\gamma}{\gamma}} + C_6 \left(\log\frac{3B}{\epsilon} + \log\log\frac{3B}{\epsilon} + C_7\right)^{\frac{\gamma+2}{\gamma}}, , & (Exponential\ Decay) \\ C_5 \left(\frac{3R}{\epsilon}\right)^{\frac{2}{\gamma-1}} \log\left(1 + \frac{12R}{\epsilon}\right) + C_8 \left(\frac{3B}{\epsilon}\right)^{\frac{4}{\gamma-1}} \left(\log\frac{3B}{\epsilon} + C_9\right), & (Polynomial\ Decay) \end{cases}$$
$$+ \log\left(1 + \frac{3B}{\epsilon\sqrt{\lambda}}\right).$$

The proof is complete. $\qquad\square$

### C.8 Proof of Lemma 7

*Proof.* We observe $r_h^n + \widehat{V}_{h+1}$ uniformly bounded by $H$. For any $f$ with $\|f\|_{\mathcal{H}} \leq 1$, similar to the proof of Lemma 4, we have

$$\|\widehat{w}_h\|_{\mathcal{H}} = \sup_{\|f\|_{\mathcal{H}} \leq 1} \langle f, \widehat{w}_h \rangle$$
$$= \sup_{\|f\|_{\mathcal{H}} \leq 1} \left\langle f, \Sigma_h^{-1} \Phi_h^\top [r_h^1 + \widehat{V}_{h+1}(\widehat{\omega}_{h+1}^1), \dots, r_h^N + \widehat{V}_{h+1}(\widehat{\omega}_{h+1}^N)]^\top \right\rangle$$
$$\leq \sup_{\|f\|_{\mathcal{H}} \leq 1} H\lambda^{-1/2} \|f\|_{\mathcal{H}} \sqrt{\sum_{i=1}^N \left\langle \mu(\widehat{\omega}_h^n, a_h^n), \Sigma_h^{-1}\mu(\widehat{\omega}_h^n, a_h^n) \right\rangle}$$
$$\leq H\lambda^{-1/2}\sqrt{\log\det(I + K_h/\lambda)},$$

where the second last inequality follows from Cauchy-Schwarz inequality and the last inequality invokes Lemma 11 in Abbasi-Yadkori et al. (2011). $\qquad\square$

## D Technical Results in RKHS

### D.1 Covering Number of RKHS

**Lemma 8** (Restatement of Lemma D.2 in Yang et al. (2020b)). Suppose Assumptions 1 and 2 hold. Let $\mathcal{H}_K(R) = \{f \in \mathcal{H}_K : \|f\|_{\mathcal{H}} \leq R\}$ be a norm ball in the reproducing kernel Hilbert space $\mathcal{H}_K$. For any $\epsilon > 0$, the covering number $\mathcal{N}(\epsilon, \mathcal{H}_K(R), \|\cdot\|_\infty)$ is bounded by

$$\log \mathcal{N}(\epsilon, \mathcal{H}_K(R), \|\cdot\|_\infty) \leq \begin{cases} \gamma \log(1 + 2R/\epsilon), & (Finite\ Spectrum) \\ C_3 \left(\log(R/\epsilon) + C_4\right)^{1+1/\gamma}, & (Exponential\ Decay) \, , \\ C_5(R/\epsilon)^{2/(\gamma-1)} \log(1 + 4R/\epsilon), & (Polynomial\ Decay) \end{cases}$$

where $C_3, C_4$ are positive constants depending on $C_1, C_2, \log R$, and constant $C_5$ depends on $C, \log R, \gamma$.

*Proof.* The idea is to transform the covering of $\mathcal{H}_K(R)$ to a proper covering on a Euclidean ball, whose dimension is determined by the spectrum of kernel $K$.

**Case 1**: (*Finite Spectrum*). For any function $f \in \mathcal{H}_K(R)$, we have

$$f = \sum_{i=1}^\gamma w_i \sqrt{\sigma_i} \nu_i \quad \text{with} \quad \sum_{i=1}^\gamma w_i^2 \leq R^2.$$

Consider two functions $f = \sum_{i=1}^{\gamma} w_i \sqrt{\sigma_i} \nu_i$ and $f' = \sum_{i=1}^{\gamma} w'_i \sqrt{\sigma_i} \nu_i$ satisfying $\sum_{i=1}^{\gamma} (w_i - w'_i)^2 \le \epsilon_0^2$. Then we have

$$
\begin{aligned}
\|f - f'\|_\infty &= \left\| \sum_{i=1}^{\gamma} (w_i - w'_i) \sqrt{\sigma_i} \nu_i \right\|_\infty \\
&\overset{(i)}{\le} \sup_\mu \sqrt{\sum_{i=1}^{\gamma} \sigma_i \nu_i^2(\mu)} \sqrt{\sum_{i=1}^{\gamma} (w_i - w'_i)^2} \\
&= \sqrt{\sup_\mu K(\mu, \mu) \cdot \sum_{i=1}^{\gamma} (w_i - w'_i)^2} \\
&\le \epsilon_0,
\end{aligned}
$$

where inequality $(i)$ uses the Cauchy-Schwarz inequality. The above inequality establishes the equivalence between an $\epsilon_0$-covering on $\mathcal{B}^\gamma(R) = \{w \in \mathbb{R}^\gamma : \|w\|_2 \le R\}$ and an $\epsilon_0$-covering on $\mathcal{H}_K(R)$. By the volume ratio argument, we know

$$
\mathcal{N}(\epsilon, \mathcal{B}^\gamma(R), \|\cdot\|_2) \le (1 + 2R/\epsilon)^\gamma.
$$

Therefore, we immediately obtain

$$
\log \mathcal{N}(\epsilon, \mathcal{H}_K(R), \|\cdot\|_\infty) \le \gamma \log(1 + 2R/\epsilon).
$$

**Case 2**: (*Exponential Decay*). In both **Case 2** and **Case 3**, we properly truncate the spectrum of kernel $K$, which reduces to **Case 1**. Thanks to the specific spectrum decay, we are able to estimate the truncation error. Specifically, let $M$ be a positive integer to be determined. We denote $\Pi_M$ as the projection operator onto the eigenspace spanned by $\nu_1, \ldots, \nu_M$. Now for any $f \in \mathcal{H}_K(R)$ and $M \ge \left( \frac{1-\gamma}{C_2 \gamma} \right)^{1/\gamma}$ for $\gamma \in (0, 1)$, we bound

$$
\begin{aligned}
\|f - \Pi_M f\|_\infty &= \left\| \sum_{i=M+1}^{\infty} w_i \sqrt{\sigma_i} \nu_i \right\|_\infty \\
&\le \sqrt{\sum_{i=M+1}^{\infty} w_i^2} \sqrt{\sup_\mu \sum_{i=M+1}^{\infty} \sigma_i \nu_i^2(\mu)} \\
&\overset{(i)}{\le} R \sqrt{\sum_{i=M+1}^{\infty} C_1 \exp(-C_2 i^\gamma)} \\
&\overset{(ii)}{\le} R \sqrt{\int_M^\infty C_1 \exp(-C_2 x^\gamma) \, dx} \\
&\overset{(iii)}{\le} \begin{cases} R \sqrt{\frac{C_1}{C_2} \exp(-C_2 M^\gamma)} & \gamma \ge 1 \\ R \sqrt{\frac{C_1 M^{1-\gamma}}{C_2 \gamma} \exp(-C_2 M^\gamma)} & \gamma \in (0, 1) \end{cases}.
\end{aligned}
\tag{30}
$$

where inequality $(i)$ follows from the uniform upper bound on $\nu_i$, inequality $(ii)$ uses the monotonicity of exponential function, and inequality $(iii)$ is valid due to the following computation. For $\gamma \ge 1$, we derive

$$
\int_M^\infty C_1 \exp(-C_2 x^\gamma) \, dx \le \int_M^\infty C_1 x^{\gamma-1} \exp(-C_2 x^\gamma) \, dx = \frac{C_1}{C_2} \exp(-C_2 M^\gamma).
$$

When $\gamma \in (0,1)$, and $M$ sufficiently large, we can also bound the truncation error using a slightly more complicated argument. Indeed, using integration by parts, we have

$$\int_M^\infty C_1 \exp\left(-C_2 x^\gamma\right) dx$$

$$= \int_{M^\gamma}^\infty C_1 \exp(-C_2 u) \frac{1}{\gamma} u^{1/\gamma - 1} du$$

$$= \frac{C_1}{C_2 \gamma} M^{1-\gamma} \exp(-C_2 M^\gamma) + \frac{C_1}{C_2} \int_{M^\gamma}^\infty \frac{1}{\gamma} \left(\frac{1}{\gamma} - 1\right) u^{1/\gamma - 2} \exp(-C_2 u) du$$

$$\leq \frac{C_1}{C_2 \gamma} M^{1-\gamma} \exp(-C_2 M^\gamma) + \frac{C_1}{C_2 M^\gamma} \int_{M^\gamma}^\infty \frac{1}{\gamma} \left(\frac{1}{\gamma} - 1\right) u^{1/\gamma - 1} \exp(-C_2 u) du$$

$$= \frac{C_1}{C_2 \gamma} M^{1-\gamma} \exp(-C_2 M^\gamma) + \frac{1}{C_2 M^\gamma} \left(\frac{1}{\gamma} - 1\right) \int_M^\infty C_1 \exp\left(-C_2 x^\gamma\right) dx,$$

which implies, for $M \geq \left(\frac{1-\gamma}{C_2 \gamma}\right)^{1/\gamma}$,

$$\int_M^\infty C_1 \exp\left(-C_2 x^\gamma\right) dx \leq \frac{C_1 M^{1-\gamma}}{C_2 \gamma - (1-\gamma) M^{-\gamma}} \exp(-C_2 M^\gamma)$$

$$\leq \frac{C_1 M^{1-\gamma}}{C_2 \gamma} \exp(-C_2 M^\gamma).$$

To this end, we construct an $\epsilon_0$-covering on $\Pi_M \mathcal{H}_K(R) = \{\Pi_M f : f \in \mathcal{H}_K(R)\}$, whose covering number is given in **Case 1**,

$$\mathcal{N}(\epsilon_0, \Pi_M \mathcal{H}_K(R), \|\cdot\|_\infty) \leq (1 + 2R/\epsilon_0)^M.$$

For a given $\epsilon$, and a proper absolute constant $c_0$, we choose $M$ to be the smallest integer satisfying

$$R\sqrt{\int_M^\infty C_1 \exp\left(-C_2 x^\gamma\right) dx} \leq \epsilon/2 \quad \implies \quad M = c_0 \left\lceil \left(\frac{2}{C_2} \log \frac{2R}{\epsilon} + \frac{1}{C_2} \log \frac{C_1}{C_2}\right)^{1/\gamma} \right\rceil \quad (31)$$

and $\epsilon_0 = \epsilon/2$. We claim that the $\epsilon/2$-covering of $\Pi_M \mathcal{H}_K(R)$ with $M$ chosen in (31) is also an $\epsilon$-covering of $\mathcal{H}_K(R)$. To see this, for any given $f \in \mathcal{H}_K(R)$, there exists $\bar{f}_M$ in the covering of $\Pi_M \mathcal{H}_K(R)$ such that $\left\|\bar{f}_M - \Pi_M f\right\|_\infty \leq \epsilon/2$. Then we have

$$\left\|f - \bar{f}_M\right\|_\infty \leq \left\|f - \Pi_M f\right\|_\infty + \left\|\Pi_M f - \bar{f}_M\right\|_\infty \leq \epsilon.$$

This implies the $\epsilon$-covering number of $\mathcal{H}_K(R)$ is

$$\log \mathcal{N}(\epsilon, \mathcal{H}_K(R), \|\cdot\|_\infty) \leq M \log(1 + 4R/\epsilon)$$

$$= c_0 \left\lceil \left(\frac{2}{C_2} \log \frac{2R}{\epsilon} + \frac{1}{C_2} \log \frac{C_1}{C_2}\right)^{1/\gamma} \right\rceil \log(1 + 4R/\epsilon)$$

$$= C_3 \left(\log(R/\epsilon) + C_4\right)^{1+1/\gamma},$$

where constants $C_3, C_4$ depend on $C_1, C_2, R$.

**Case 3**: (*Polynomial Decay*). The idea is the same as in **Case 2**, except a different upper bound on the truncation error. Specifically, for polynomial decay spectrum and any $f \in \mathcal{H}_K(R)$, we have

$$\|f - \Pi_M f\|_\infty = \left\| \sum_{i=M+1}^\infty w_i \sqrt{\sigma_i} \nu_i \right\|_\infty$$

$$\leq R\sqrt{\int_M^\infty C x^{-\gamma} dx}$$

$$\leq R\sqrt{(\gamma - 1)^{-1} C M^{-\gamma + 1}}. \quad (32)$$

By letting $M$ be the smallest integer satisfying $R\sqrt{(\gamma-1)^{-1}CM^{-\gamma+1}} \le \epsilon/2$, i.e.,

$$M = \left\lceil \left(\frac{4R^2C}{\epsilon^2(\gamma-1)}\right)^{\frac{1}{\gamma-1}} \right\rceil,$$

we have

$$\log \mathcal{N}(\epsilon, \mathcal{H}_K(R), \|\cdot\|_\infty) \le M \log(1+4R/\epsilon) = \left\lceil \left(\frac{4R^2C}{\epsilon^2(\gamma-1)}\right)^{\frac{1}{\gamma-1}} \right\rceil \log(1+4R/\epsilon)$$

$$= C_5 \left(\frac{R}{\epsilon}\right)^{\frac{2}{\gamma-1}} \log(1+4R/\epsilon),$$

where constant $C_5$ depends on $C, R, \gamma$. The proof is complete. $\qquad\square$

**Lemma 9.** Suppose Assumption 1 and 2 hold. Let $\mathcal{F}_K(\lambda) = \{\Sigma : \|\Sigma\|_{\mathrm{op}} \overset{\triangle}{=} \sup_{\|f\|_{\mathcal{H}}\le 1} \langle f, \Sigma f\rangle_{\mathcal{H}} \le \lambda^{-1}\}$ be a collection of operators of bounded operator norm defined on the RKHS $\mathcal{H}_K$. For any $\epsilon > 0$, the covering number $\mathcal{N}(\epsilon, \mathcal{F}_K(\lambda), \|\cdot\|_{\mathrm{op}})$ is bounded by

$$\log \mathcal{N}(\epsilon, \mathcal{F}_K(\lambda), \|\cdot\|_{\mathrm{op}}) \le \begin{cases} \gamma^2 \log(1+2\sqrt{\gamma}/(\lambda\epsilon)), & (Finite\ Spectrum) \\ C_6 \left(\log(1/\epsilon) + \log\log(1/\epsilon) + C_7\right)^{1+2/\gamma}, & (Exponential\ Decay) \\ C_8(1/\epsilon)^{4/(\gamma-1)} \left(\log(1/\epsilon) + C_9\right), & (Polynomial\ Decay) \end{cases},$$

where $C_6, C_7$ are positive constants depending on $C_1, C_2, \lambda$, and constants $C_8, C_9$ depend on $C, \lambda, \gamma$.

*Proof.* Similar to the proof of Lemma 8, the idea here is to transform the covering of $\mathcal{F}_K(\lambda)$ to a proper matrix covering, whose dimension is determined by the spectrum of kernel $K$.

**Case 1**: (*Finite Spectrum*). We show an equivalence between operator $\Sigma$ and square matrix $M \in \mathbb{R}^{\gamma\times\gamma}$. For any unit norm eigenfunction $\sqrt{\sigma_i}\nu_i \in \mathcal{H}_K$, we denote

$$\Sigma(\sqrt{\sigma_i}\nu_i) = \sum_{j=1}^{\gamma} w_{ij}\sqrt{\sigma_j}\nu_j.$$

Let matrix $W_{ij} = w_{ij}$. Then for any $f \in \mathcal{H}_K$, we write $f = \sum_{j=1}^{\gamma} a_j \sqrt{\sigma_j}\nu_j$. Some algebra gives rise to

$$\langle f, \Sigma f\rangle_{\mathcal{H}} = a^\top W a \quad \text{with} \quad a = [a_1, \ldots, a_\gamma]^\top.$$

This yields a one-to-one correspondence between $\mathcal{F}_K(\lambda)$ and $\mathcal{W}_\gamma(\lambda) = \{W \in \mathbb{R}^{\gamma\times\gamma} : \lambda^{-1}I \succeq W \succeq 0\}$. Therefore, it suffices to find the covering number of $\mathcal{W}_\gamma(\lambda)$. By vectorize a $\gamma$-by-$\gamma$ matrix as a $\gamma^2$-dimensional vector, we obtain the covering number of $\mathcal{W}_\gamma(\lambda)$ using the volume ratio argument:

$$\log \mathcal{N}(\epsilon, \mathcal{W}_\gamma(\lambda), \|\cdot\|_{\mathrm{F}}) \le \gamma^2 \log(1+2\sqrt{\gamma}/(\lambda\epsilon)),$$

where $\|\cdot\|_{\mathrm{F}}$ denotes the Frobenius norm. Accordingly, we have

$$\log \mathcal{N}(\epsilon, \mathcal{F}_K(\lambda), \|\cdot\|_{\mathrm{op}}) \le \gamma^2 \log(1+2\sqrt{\gamma}/(\lambda\epsilon)).$$

**Case 2**: (*Exponential Decay*). We truncate the spectrum of kernel $K$ again. Let $M$ be a positive integer to be determined. We denote $\Pi_M$ as the projection operator onto the eigenspace spanned by $\nu_1, \ldots, \nu_M$. For any $f \in \mathcal{H}_K$ with $\|f\|_{\mathcal{H}} \le 1$, the truncation error is bounded by (30):

$$\|f - \Pi_M f\|_\infty \le \begin{cases} \sqrt{\frac{C_1}{C_2}\exp(-C_2M^\gamma)} & \gamma \ge 1 \\ \sqrt{\frac{C_1M^{1-\gamma}}{C_2\gamma}\exp(-C_2M^\gamma)} & \gamma \in (0,1) \end{cases}.$$

By the linearity, we have

$$\langle f, \Sigma f\rangle = \langle \Pi_M f, \Sigma(\Pi_M f)\rangle + \langle f - \Pi_M f, \Sigma(\Pi_M f)\rangle + \langle f, \Sigma(f - \Pi_M f)\rangle. \qquad (33)$$

The last two terms in (33) can be bounded by the truncation error:

$$\langle f - \Pi_M f, \Sigma \Pi_M f \rangle + \langle f, \Sigma(f - \Pi_M f) \rangle \leq 2\lambda^{-1} \begin{cases} \sqrt{\frac{C_1}{C_2} \exp(-C_2 M^\gamma)} & \gamma \geq 1 \\ \sqrt{\frac{C_1 M^{1-\gamma}}{C_2 \gamma} \exp(-C_2 M^\gamma)} & \gamma \in (0,1) \end{cases}.$$

Given an $\epsilon > 0$, by choosing $M$ to be the smallest integer satisfying

$$2\lambda^{-1} \sqrt{\frac{C_1}{C_2} \exp(-C_2 M^\gamma)} \leq \epsilon/2 \implies M = c_0 \left\lceil \left( \frac{2}{C_2} \log \frac{4}{\lambda\epsilon} + \frac{1}{C_2} \log \frac{C_1}{C_2} \right)^{1/\gamma} \right\rceil, \quad (34)$$

for some absolute constant $c_0$, we only need to find an $\epsilon/2$-covering of $\Sigma$ evaluated on $\Pi_M \mathcal{H}_K$. This reduces to **Case 1**, and the covering number is bounded by $(1 + 2\sqrt{M}/(\lambda\epsilon))^{M^2}$. With the choice of $M$ in (34), we obtain

$$\log \mathcal{N}(\epsilon, \mathcal{F}_K(\lambda), \|\cdot\|_{\mathrm{op}}) \leq M^2 \log(1 + 4\sqrt{M}/(\lambda\epsilon))$$

$$= c_0 \left\lceil \left( \frac{2}{C_2} \log \frac{4}{\lambda\epsilon} + \frac{1}{C_2} \log \frac{C_1}{C_2} \right)^{1/\gamma} \right\rceil^2 \log(1 + 4\sqrt{M}/(\lambda\epsilon))$$

$$= C_6 \left( \log \left( \frac{\log 1/\epsilon}{\epsilon} \right) + C_7 \right)^{1+2/\gamma},$$

where constants $C_6, C_7$ depend on $C_1, C_2, \lambda$.

**Case 3**: (*Polynomial Decay*). The idea is the same as in **Case 2**. For polynomial decay spectrum and any $f \in \mathcal{H}_K$ with $\|f\|_{\mathcal{H}} \leq 1$, by (32), the truncation error is bounded by

$$\|f - \Pi_M f\|_\infty \leq \sqrt{(\gamma - 1)^{-1} C M^{-\gamma+1}}.$$

By letting $M$ be the smallest integer satisfying $2\sqrt{(\gamma - 1)^{-1} C M^{-\gamma+1}} \leq \epsilon/2$, i.e.,

$$M = \left\lceil \left( \frac{16 C}{\epsilon^2 (\gamma - 1)} \right)^{\frac{1}{\gamma - 1}} \right\rceil,$$

we have

$$\log \mathcal{N}(\epsilon, \mathcal{F}_K(\lambda), \|\cdot\|_{\mathrm{op}}) \leq M^2 \log(1 + 4\sqrt{M}/(\lambda\epsilon)) = \left\lceil \left( \frac{16 C}{\epsilon^2 (\gamma - 1)} \right)^{\frac{1}{\gamma - 1}} \right\rceil^2 \log(1 + 4\sqrt{M}/(\lambda\epsilon))$$

$$= C_8 \left( \frac{1}{\epsilon} \right)^{\frac{4}{\gamma - 1}} (\log(1/\epsilon) + C_9),$$

where constants $C_8, C_9$ depend on $C, \lambda, \gamma$. The proof is complete. $\qquad \square$

### D.2 Effective Dimension of RKHS

**Lemma 10.** Suppose that Assumptions 1 and 2 hold. Recall that the Gram matrix $[K_h]_{\ell,\ell'} = K((\widehat{\omega}_h^\ell, a_h^\ell), (\widehat{\omega}_h^{\ell'}, a_h^{\ell'}))$ defined on dataset $\mathcal{D}_{N,H}$. For any $h = 1, \ldots, H$ and fixed $\lambda > 0$, we have

$$\log \det(I + K_h/\lambda) \leq \begin{cases} C_{\text{eff-FS}} \cdot \max\{d, \gamma\} \log N & (\textit{Finite Spectrum}) \\ C_{\text{eff-ED}} \cdot d(\log N)^{1+1/\gamma} & (\textit{Exponential Decay}) , \\ C_{\text{eff-PD}} \cdot d N^{\frac{d+1}{\gamma+d}} \log N & (\textit{Polynomial Decay}) \end{cases}$$

where $d$ is the dimension of mete state-action space $\mathcal{S} \times \mathcal{S} \times \mathcal{A}$, constant $C_{\text{eff-FS}}$ depends $\lambda$ and Lebesgue measure of meta state-action space $\Xi$, constant $C_{\text{eff-ED}}$ depends on $C_1, C_2, \lambda$, and Lebesgue measure of meta state-action space $\Xi$, and constant $C_{\text{eff-PD}}$ depends on $C, \lambda, \gamma$, and Lebesgue measure of meta state-action space $\Xi$.

*Proof.* Each entry in the Gram matrix $K_h \in \mathbb{R}^{N \times N}$ can be written as $[K_h]_{\ell,\ell'} = \mathbb{E}_{s_1 \sim \widehat{d}_h^{\ell}, s_2 \sim \widehat{d}_h^{\ell'}}[K((s_{h,0}^{\ell}, s_1, a_h^{\ell}), (s_{h,0}^{\ell'}, s_2, a_h^{\ell'}))]$. Correspondingly, we define $\widetilde{K}_h \in \mathbb{R}^{N \times N}$ as $[\widetilde{K}_h]_{\ell,\ell'} = K((s_{h,0}^{\ell}, s_{h,1}^{\ell}, a_h^{\ell}), (s_{h,0}^{\ell'}, s_{h,1}^{\ell'}, a_h^{\ell'}))$, which can be viewed as only sampling a single agent from the mean-field state distribution. Observe that $\log \det$ is a concave function. By Jensen's inequality, we have

$$\log \det(I + K_h/\lambda) \leq \mathbb{E}[\log \det(I + \widetilde{K}_h/\lambda)] \leq \sup_{\widetilde{K}_h} \log \det(I + \widetilde{K}_h/\lambda),$$

where the expectation is taken over the empirical states of $m$ agents. To bound $\log \det(I + K_h/\lambda)$, we only need to bound $\sup_{\widetilde{K}_h} \log \det(I + \widetilde{K}_h/\lambda)$. We introduce several notations. For fixed $\tau > 0$, we denote $C_\tau = 2\mu(\Xi)(2\tau + 1)$ with $\mu(\Xi)$ being the Lebesgue measure of $\Xi$ and $n_\tau = N^\tau \log N$. Moreover, we denote $d$ as the dimension of $\Xi$ and $B_\sigma(N_\star) = \sum_{i=N_\star+1}^{\infty} \sigma_i$ as the tail spectrum of kernel $K$.

By Theorem 8 in Srinivas et al. (2009), for any positive integer $N_\star \leq n_\tau$, we have

$$\sup_{\widetilde{K}_h} \log \det(I + \widetilde{K}_h/\lambda) \leq N_\star \log(\lambda N^{1+\tau}) + C_\tau \lambda \log N(N^{\tau+1}B_\sigma(N_\star) + 1) + \mathcal{O}(N^{1-\tau/d}).$$

(35)

We choose $N_\star$ and $\tau$ according to the spectrum of kernel $K$.

**Case 1:** *(Finite Spectrum).* We set $N_\star = \gamma$, which implies $B_\sigma(N_\star) = 0$, and $\tau = d$. Plugging into (35), we have

$$\sup_{\widetilde{K}_h} \log \det(I + \widetilde{K}_h/\lambda) \leq \gamma \log(\lambda N^{1+\tau}) + C_\tau \lambda \log N + \mathcal{O}(1) = C_{\text{eff-FS}} \cdot \max\{d, \gamma\} \log N,$$

where $C_{\text{eff-FS}}$ depends on $\lambda, \mu(\Xi)$.

**Case 2:** *(Exponential Decay).* We first bound the tail spectrum $B_\sigma(N_\star)$. By the computation in (30), we have

$$B_\sigma(N_\star) \leq \begin{cases} \frac{C_1}{C_2}\exp(-C_2 N_\star^\gamma) & \gamma \geq 1 \\ \frac{C_1 N_\star^{1-\gamma}}{C_2 \gamma}\exp(-C_2 N_\star^\gamma) & \gamma \in (0,1) \end{cases}.$$

By setting $\tau = d$ again, substituting the upper bound of $B_\sigma(N_\star)$ into (35), and further choosing $N_\star = \frac{c_0}{C_2}(d+1)(\log N)^{1/\gamma}$, for some absolute constant $c_0$, we deduce

$$\sup_{\widetilde{K}_h} \log \det(I + \widetilde{K}_h/\lambda) \leq \frac{1}{C_2}(d+1)(\log N)^{1/\gamma}\log(\lambda N^{1+d}) + C_\tau \lambda \log N\left(1 + \frac{C_1}{C_2}\right) + \mathcal{O}(1)$$

$$= C_{\text{eff-ED}} \cdot d(\log N)^{1+1/\gamma},$$

where constant $C_{\text{eff-ED}}$ depends on $C_1, C_2, \lambda, \mu(\Xi)$.

**Case 3:** *(Polynomial Decay).* Similar to **Case 2**, we bound $B_\sigma(N_\star)$ using (32) as

$$B_\sigma(N_\star) \leq (\gamma-1)^{-1}CN_\star^{-\gamma+1}.$$

Substituting the upper bound of $B_\sigma(N_\star)$ into (35), we obtain

$$\sup_{\widetilde{K}_h} \log \det(I + \widetilde{K}_h/\lambda)$$

$$\leq N_\star \log(\lambda N^{1+\tau}) + C_\tau \lambda \log N\left(1 + N^{\tau+1}(\gamma-1)^{-1}CN_\star^{-\gamma+1}\right) + \mathcal{O}(N^{1-\tau/d}).$$

Choosing $N_\star = N^{(\tau+1)/\gamma}$, we deduce

$$\sup_{\widetilde{K}_h} \log \det(I + \widetilde{K}_h/\lambda)$$

$$\leq N^{(\tau+1)/\gamma}\log(\lambda N^{1+\tau}) + C_\tau \lambda \log N\left(1 + N^{(\tau+1)/\gamma}(\gamma-1)^{-1}C\right) + \mathcal{O}(N^{1-\tau/d}).$$

Now we set $\tau = \frac{\gamma-1}{1+\gamma/d}$ so that $N^{(\tau+1)/\gamma} = N^{1-\tau/d}$, which further gives rise to

$$\sup_{\widetilde{K}_h} \log \det(I + \widetilde{K}_h/\lambda)$$

$$\leq N^{\frac{d+1}{d+\gamma}} \log(\lambda N^{\frac{(d+1)\gamma}{d+\gamma}}) + C_\tau \lambda \log N \left(1 + N^{\frac{d+1}{d+\gamma}}(\gamma-1)^{-1}C\right) + \mathcal{O}(N^{\frac{d+1}{d+\gamma}})$$

$$= C_{\text{eff-PD}} \cdot dN^{\frac{d+1}{d+\gamma}} \log N,$$

where constant $C_{\text{eff-PD}}$ depends on $C, \lambda, \gamma, \mu(\Xi)$. In this case, we need to verify $N_\star \leq n_\tau$, which implies $\gamma \geq 2 + 1/d$. $\qquad\square$

## E   More Details on Experiment Implementation

We follow the implementation given by Liu et al. (2020a) for all experiments on this scenario. However, we fix the number of observable agents and landmarks, $k$, to 2, resulting in an observation space of 14. The environment then uses a discrete action space, corresponding to [NO-OP, LEFT, RIGHT, UP, DOWN], with movement in each direction being applied as a noisy force on the agent.

Note that we discretize the state space before episodes are used to train SAFARI by truncating continuous measurements to three significant figures. This allows us to sample fewer trajectories and shorten the length of the dataset, which in turn reduces the training time of the algorithm. As evidenced by Figure 2, performance of SAFARI is still comparable to the "expert" COMA even with a lower granularity in the representation of the state. We use the normal Gaussian kernel for the implementation here.

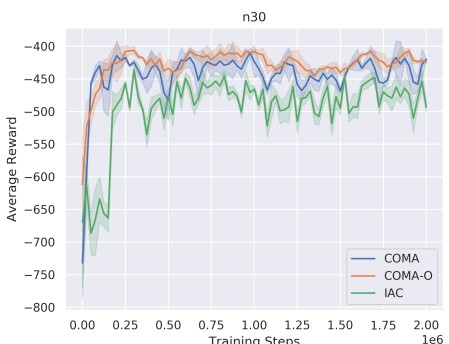

We provide in Figure 3 the performance of COMA, COMA-O, and IAC on 20 evaluation episodes during training in the 30 agent environment. The same trend follows as with the 15 agent environment: COMA-O performs slight worse than COMA but still clearly better than IAC. The final convergence reward is also reflected in Figure 2. For the 100 agent case, we use the policies trained for 15 agents (a single network due to parameter sharing), as the exponential growth of the state-action space has greatly reduced the sample efficiency of COMA and significantly increased training time.

Figure 3: Training reward on the 30 agent environment.

**Hyperparameters**   All hyperparameters are tuned by logarithmic random search over the ranges given below:

Table 1: Algorithm Hyperparameters

| Parameters | Value |
| --- | --- |
| **COMA, VDN** | |
| Discount $\gamma$ | 0.95 |
| Exploration Rate | $(0.1, 0.3)$ |
| Exploration Anneal | 0.998 |
| Policy learning rate | $(0.0001, 0.01)$ |
| Critic learning rate | $(0.001, 0.01)$ |
| Optimizer | Adam |
| **SAFARI** | |
| $\beta$ | $(0.1, 1)$ |
| $\lambda$ | $(0.1, 0.99)$ |