# OpenReview forum: "Pessimism Meets Invariance: Provably Efficient Offline Mean-Field Multi-Agent RL"
_NeurIPS.cc/2021/Conference — NeurIPS 2021 Poster_

### Official Review · Reviewer_Jqk7 · 2021-07-12

**Rating:** 8
**Confidence:** 4

**Summary:**

The paper (mainly) provides a pessimism based offline RL algorithm for MF-MARL. It extends existing literature by
1. Generalizing pessimism based offline RL algorithms from single agent to MF-MARL
2. Considering MF-MARL from an offline regime
It also extends existing literature on online MF-MARL by converting the pessimism penalty to a optimistic bonus and showed the sample efficiency of the algorithm under this regime.

**Limitations And Societal Impact:**

Limitations: assumptions of the paper are clearly stated and the paper is entirely devoted to MF-MARL.

Societal impact: purely theoretical & no direct societal impact.

**Main Review:**

- Originality: the key ideas behind SAFARI, as the authors mention in the paper, were first discussed in [1] (reference [70] in the paper itself) and the proof structure of theorem 1 largely follows the same paper. While MF-MARL is an interesting area and analyzing it from an offline perspective is indeed interesting, the proposed algorithm largely follows existing research on offline learning of linear MDPs.

- Quality: theorems and claims are well supported by theory, which are meticulously proven in the appendix.

- Clarity: well written paper with thoroughly explained proofs and very few (if any) grammatical errors. Easy to follow and clearly explained.

- Significance: it should be noted that some benefits of the algorithm (e.g. no curse of multiple agents) are caused by the nature of the MF-MARL problem, rather than the proposed algorithm. Additionally, unlike the aforementioned [1], the paper does not demonstrate the sample efficiency of the algorithm in the offline regime (of course its efficiency in the online regime is well-demonstrated and should be somewhat indicative of the offline regime). However, combining MF-MARL with PEVI is an interesting idea and, especially in the context of social welfare optimization, offline RL algorithms are appreciated and necessary.

[1] Ying Jin, Zhuoran Yang, and Zhaoran Wang. Is pessimism provably efficient for offline RL?525arXiv preprint arXiv:2012.15085, 2020.

**Time Spent Reviewing:**

2.5 hours

---

> ### Author Response · Authors · 2021-08-10
> **Author Response to Reviewer Jqk7**
>
> Thank you for your encouraging review and valuable comments.
>
> **Q1**: Originality: the key ideas behind SAFARI, as the authors mention in the paper, were first discussed in [1] (reference [70] in the paper itself) and the proof structure of theorem 1 largely follows the same paper. While MF-MARL is an interesting area and analyzing it from an offline perspective is indeed interesting, the proposed algorithm largely follows existing research on offline learning of linear MDPs.
>
> **A**: Despite the similar pessimistic value iteration framework, our paper exhibits novelty in the *multi-agent setting* and *theoretical analysis*. In particular, different from the single agent setting, we consider the off-line multi-agent setting with a mean-field state for modeling the interaction between the representative agent and other agents in the environment. This brings the challenge that the mean-field state space is infinite-dimensional. We resort to kernel embedding to address the challenge. From a theoretical perspective, our analysis explicitly establishes the covariate concentration error of mean embedding (see Line 296 in Technical Overview). Such an error stems from the fact that in practice, only finitely many agents are observed. We further provide new analysis on the uncertainty quantification for estimating the Bellman operator under covariate concentration error ($\omega$ v.s. $\hat{\omega}$), which is addressed in Bounding Term (A) in the proof of Theorem 1 (beginning from Line 630).
>
> **Q2**: Significance: it should be noted that some benefits of the algorithm (e.g. no curse of multiple agents) are caused by the nature of the MF-MARL problem, rather than the proposed algorithm. Additionally, unlike the aforementioned [1], the paper does not demonstrate the sample efficiency of the algorithm in the offline regime (of course its efficiency in the online regime is well-demonstrated and should be somewhat indicative of the offline regime). However, combining MF-MARL with PEVI is an interesting idea and, especially in the context of social welfare optimization, offline RL algorithms are appreciated and necessary.
>
> **A**: We agree that no curse of many agents stems from the permutation invariance of the MF-MARL problem itself, not the algorithm proposed in the paper.
>
> We provide an explicit sub-optimality gap in Corollary 1 under the weak coverage assumption. Such an assumption is necessary for obtaining a concrete convergence guarantee indicating the sample efficiency of the algorithm (See the response to **Q1** of Reviewer WSQ9). The convergence rate is $\tilde{O}(H^2 d_{\rm eff} / \sqrt{N})$. It is also interesting to establish a minimax lower bound in the offline setting, yet it is beyond the scope of the paper and we leave it as a future direction.

---

### Official Review · Reviewer_fQxx · 2021-07-16

**Rating:** 7
**Confidence:** 4

**Summary:**

This paper studies the Mean-Field Multi-Agent RL problem in the offline regime where pre-collected data is available. Under the assumption that data provides “sufficient” information regarding the optimal policy, the authors propose SAFARI, a pessimistic algorithm, and prove sample complexity guarantees to achieve a certain sub-optimality gap. The obtained data-dependent guarantees do not depend on the (large) number of agents and scale linearly with the effective dimension of the function class (used to parametrize the value function). Moreover, the authors propose another algorithm OMPPO for the online setting and obtain similar sublinear (in $N$; for some kernels) regret guarantees.

**Limitations And Societal Impact:**

Yes.

**Main Review:**

Recently, both offline RL and mean-field games/RL have received a lot of attention and hence, I think that the paper is timely and considers an original idea of unifying these two problems. The need for offline RL in the mean-field MARL setting is clearly motivated in the paper. The obtained theoretical results are sound. The paper is also clearly written and the related work section nicely covers the existing literature. It would be great to see the performance of the proposed algorithms in simulations and experiments, but since this is a theoretical paper, I understand why this was not done.

I like how the mean embeddings come into play when parametrizing the reward and transition (also, later on, when concentration results for empirical mean embedding are invoked in the analysis). By having a feature mapping that depends on $\mathcal{S} \times \mathcal{S} \times \mathcal{A}$, it is very convenient to invoke upper bounds on $d_{\text{eff}}$ in the final results. However, I feel that this parametrization also has its disadvantages,  as under such parametrization, the reward function (also transition) is only averaged with respect to $d_s$, i.e., the dependency on $d_s$ is quite simple and can be thought of as averaging over a single contextual parameter. This is reflected in the main bounds where $d$ corresponds to the dimension of the $S \times S \times A$ (thinking of it in this way, it appears that the problem/algorithm is not much different than the single-agent setup with a contextual variable (or two agent setting), which is also reflected in the bounds). Alternatively, one can consider having a kernel that is defined over $\mathcal{M}(S)$ instead of $S$ (i.e., $S \times \mathcal{M}(S) \times A$), and allow for rewards and transitions that depend on $d_s$ in more complex ways. I’m wondering if the authors can briefly comment on this.


Can you comment on the assumption on line 238 that involves $v_h(x)$ and provide the intuition behind it? This assumption seems like an atypical one to me. How does it impact the analysis?

It’d be great if the authors can comment on some of the implementation details of SAFARI. In particular, solving for the optimal policy $\hat{\pi_h}$ when the set of actions is infinite. Moreover, if I understood correctly (since it is not explicitly written in the algorithm) when searching for the optimal policy $\hat{\pi}_h$, one needs to solve the previous argmax problem for each $w \in \Omega$? How should this be done in practice given that $\Omega$ contains $\mathcal{M}(S)$ (i.e., space of probability distributions)?

In summary, I think this paper considers an interesting problem, provides further insights into the mean-field MARL, and contains some nice theoretical results/algorithms.

Minor comments:

-- Line 15-16: “we also propose a variant of our SAFARI algorithm”; I would argue that this is a new algorithm rather than a variant of SAFARI.

-- Line 140: $d_{h+1,j}$; Should it be $s_{h+1,j}$ instead?

-- Line 77: “these methods focus -on- the online setting”

-- Line 304: “we denote $s_0$ as the representative agent”

-- Line 196: “gram” -> “Gram”

-- Equation 5: $\alpha?$


**Time Spent Reviewing:**

4-5

---

> ### Author Response · Authors · 2021-08-10
> **Author response to Reviewer fQxx**
>
> Thank you for your careful review and valuable comments.
>
> **Q1**: Alternatively, one can consider having a kernel that is defined over $\mathcal{M}(\mathcal{S})$ instead of $\mathcal{S}$, and allow for rewards and transitions that depend on $d_s$ in more complex ways. I’m wondering if the authors can briefly comment on this.
>
> **A**: We consider a kernel defined on $\mathcal{S}$ for simplicity. However, our analysis can be extended to kernels defined over $\mathcal{M}(\mathcal{S})$. In detail, we can first embed the meta state space $\Omega$ into an RKHS induced by a kernel $k(\cdot, \cdot)$. We define the mean embedding $\mu_\zeta$ as $\mu_{\zeta}(\cdot) = \int_{\Omega} k(\cdot, t) d \zeta(t)$ for $\zeta = p_\omega \times p_a$ being a measure on $\Omega \times \mathcal{A}$. Given the mean embedding $\mu_{\zeta}$, we can parameterize the reward and transition probabilities in another RKHS induced by kernel $K(\cdot, \cdot)$. As can be seen, having a kernel defined over $\mathcal{M}(\mathcal{S})$ boils down to replacing the embedding $\mu(\omega, a)$ in Equation (3) by its counterpart $\mu_{\zeta}$. To this end, our analysis can be readily applied (the dimension $d$ of $\mathcal{S} \times \mathcal{S} \times \mathcal{A}$ should be replaced by the corresponding effective dimension of RKHS induced by kernel $k$).
>
> **Q2**: Can you comment on the assumption on Line 238 that involves $v_h(x)$ and provide the intuition behind it.
>
> **A**: Assumption on Line 238, $\int \lVert v_h\rVert_{\mathcal{H}} \leq \sqrt{d_{\rm eff}}$, is much weaker compared to $\lVert v_h \rVert_{\mathcal{H}}$ being bounded. The latter is a common technical assumption for imposing boundedness in the transition probabilities. Our assumption on the norm of $v_h$ translates to a bound on the Bellman operator and indicates the difficulty of estimating it. Specifically, in Bounding Term $(B)$ in the proof of Theorem 1, Equation (15) shows that $(B)$ is proportional to the RKHS norm $\int \lVert v_h \rVert_{\mathcal{H}}$.
>
> **Q3**: It would be great if the authors can comment on some of the implementation details of SAFARI.
>
> **A**: Our algorithm does not need to enumerate each $\omega \in \Omega$ for solving the argmax problem. Once we randomly sample and fix a starting state $\omega$, the SAFARI algorithm returns the learned policy $\hat{\pi}$ and estimated reward $V_1^{\hat{\pi}}(\omega)$. During each iteration the algorithm, we need to estimate value function $\hat{V}_h$ evaluated at $\hat{\omega_h^n}$ for $n = 1, \dots, N$ and $h = 2, \dots, H$, as indicated by Equation (6) for Bellman operator estimation. In the last step $h=1$, we need to return the estimated value function evaluated at the initial state $\omega$. Therefore, we do not need to solve the argmax in the policy optimization step for each $\omega$ in an infinite-dimensional meta state space.
>
> In the case of continuous action, solving for the optimal policy needs to solve a continuous optimization problem. We may use gradient descent algorithms to solve the argmax problem. Alternatively, we may discretize the continuous action space, and search an approximated optimal policy.
>
> **Q4**: Minor comments.
>
> **A**: Thank you for your careful review. We will correct typos in the next version.
>
> Line 140 should be $s_{h+1, j}$.
>
> Line 77 should be “focus on”.
>
> Equation 5, $\alpha$ should be removed, and the optimization is over $f$.

---

> > ### Comment · Reviewer_fQxx · 2021-08-28
> > **Related work [15]**
> >
> > Thank you for your response. Can you please comment on the differences/similarities with respect to [15]? It seems that this work might be the first one to introduce the idea of mean embeddings in MF-MARL and I feel that this is not properly accredited in the paper.

---

> > > ### Author Response · Authors · 2021-08-28
> > > **Re: Related work [15]**
> > >
> > > Dear Reviewers:
> > >
> > > Thank you for your feedback.
> > >
> > > [15] (Breaking the curse of many agents: Provable mean embedding q-iteration for mean-field reinforcement learning) studies a similar mean-field reinforcement learning setting and also adopts the idea of mean embedding. In the following, we provide an in-depth comparison with this work in three aspects: _modelling, assumption_, and _algorithm_.
> > >
> > > ### Modelling
> > > In terms of the (multi-agent) MDP model, [15] considers a **centralized** setting where there is a central controller that takes actions from a **finite** action space, and the state space is the mean-field state, i.e., distribution of the local states of a set of agents. Thus, their MDP model is intrinsically a _standard MDP_ with the state space being $\mathcal{M}(\mathcal{S})$, where $\mathcal{S}$ is the space of local state and $\mathcal{M} (\mathcal{S})$ denote the space of probability distributions over $\mathcal{S}$.
> > >
> > > In contrast, in our multi-agent MDP model, each agent has **local states** in $\mathcal{S}$ and are able to take **local actions** from $\mathcal{A}$. In addition, the transition of local states of each agent, say, agent $i$,  depends on the current local state $s^i$, current local action $a^i$, and the mean-field state $\mu^i$ (distribution of the local states of all the agents).
> > >
> > > It seems infeasible to transform our model into the setting of [15]. The reason is that when there is a central controller and the number of agents is $N$, the central controller needs to **take joint actions** from the joint action space $\mathcal{A}^{N}$, whose size grows exponentially in $N$. In our work, we get around such a _curse of many agents_ by exploiting the **permutation invariance** in the policy.
> > >
> > > ### Assumption on the data
> > > [15] assumes the dataset has uniform coverage over all policies, which is measured by the uniformly bounded **concentrability coefficients** (Assumption 3.1). In contrast, our coverage assumption on the data is strictly weaker. Our Theorem 1  holds for arbitrary data distribution and our Theorem 2, which establishes the statistical rates, assumes only **weak coverage**. Here weak coverage means that the dataset only contains sufficient information about the optimal policy.
> > >
> > > ### Assumption on kernels
> > > [15] require a polynomial decay of the kernel spectrum with both **upper and lower** bounds, i.e., the eigenvalue $t_n$ satisfies $\alpha \leq n^b t_n \leq \beta$ for an integer $n$ and positive constants $\alpha, \beta$ (Assumption 3.3 in [15]). While our theory holds for general spectrum assumptions on the kernel (see Assumption 2, which includes finite spectrum, exponential decay, and polynomial decay cases).
> > >
> > > ### Algorithm
> > > Although both our work and [15] utilize mean embedding in the algorithm, our algorithm is based on the **pessimism** principle, which is adopted to tackle the challenge brought by weak coverage of the dataset. In contrast, the algorithm in [15] seems a modification of the classical fitted Q iteration algorithm for distribution mean embeddings. Due to the need of achieving pessimism, in addition to estimation, we need to quantify the uncertainty of the  ridge  regression estimator, which is missing in [15].
> > >
> > >  In our submitted version, we commented on [15] by comparing the modeling aspect:
> > > > [15] investigate a mean-field MDP motivated by permutation invariance. They require a central controller managing the actions of all the agents, and therefore are restricted to handling the curse of many agents from the exponential blowup of joint state space.
> > >
> > > We agree with the reviewer that [15] first introduces mean embedding to mean-field MARL. In the revised version, we will provide a more in-depth comparison and correctly describe its contribution.

---

> > > > ### Comment · Reviewer_fQxx · 2021-08-30
> > > > **Related work [15]**
> > > >
> > > > Thank you for your detailed explanation.

---

### Official Review · Reviewer_WSQ9 · 2021-07-16

**Rating:** 6
**Confidence:** 4

**Summary:**

The paper presents a method which uses Kernel mean embedding for mean-field based MARLwith  offline settings. Under a weak data coverage assumption, it shows a suboptimality bound in offline settings and extend the result to online settings. No experiments are included in this paper.

**Main Review:**

1) I wonder if the weak data coverage assumption is too strict to implement, which means it needs to be tested in experiments. In addition, it would be better if there were comparisons with other algorithms in MARL.
2) section 5 spend a lot of time to describe various assumptions made, but there is no any experimental verifications.
3) in the online learning setting, we may have no information about the state of other agents. In such cases, what can we do?
4) for eq.5, there is an i.i.d assumption for each kernel mean embedding u, which could be problematic.



**Time Spent Reviewing:**

3

---

> ### Author Response · Authors · 2021-08-10
> **Author response to Reviewer WSQ9**
>
> Thank you for your comments.
>
> **Q1**: I wonder if the weak data coverage assumption is too strict to implement, which means it needs to be tested in experiments. In addition, it would be better if there were comparisons with other algorithms in MARL.
>
> **A**: The weak data coverage assumption is necessary for the statistical convergence results in Corollary 1. Such an assumption, however, is much weaker compared to commonly-made data coverage assumptions in existing literature (e.g., uniform lower bound on visitation measure). See more in Section 2 Line 104-106. We will make a clarification following Assumption 4 in the next version.
> Under the offline setting where we cannot interact with the environment, for standard MDP, or even the much simpler multi-arm bandit problem, certain coverage assumption on the dataset is required. For example, consider a multi-arm bandit problem in the offline setting, if the dataset does not contain observations of the optimal arm, then it seems impossible to find the optimal arm. Our weak data coverage assumption means that the dataset contains sufficient observations of the optimal arm only, while it is possible that some suboptimal arms do not have sufficient observations and thus have high uncertainty. In comparison, uniform data coverage (e.g., uniform lower bound on visitation measure) requires that the dataset contains sufficient observations of all the arms.
>
> We compare with other algorithms for MARL in Section 2. To briefly restate, our proposed offline algorithm is not directly comparable to existing algorithms, since they all focus on the online setting (see Line 77-78).
>
> **Q2**: Section 5 spends a lot of time to describe various assumptions made, but there is no experimental verification.
>
> **A**: Our assumptions in Section 5 are practice-driven and very general. For example, Assumption 1 holds for a rich family of kernels, including the most commonly used ones, RBF and Laplace kernels (Line 211-212). Assumption 2 characterizes the spectrum of kernels in three categories. These categories cover the vast majority of commonly used kernels, including RBF, Laplace, and neural tangent kernels. See more examples in Line 227-235. Moreover, our theory provides separate results for these categories. Assumption 4 is discussed in the previous question **Q1**.
>
> We will provide experiments in the next version.
>
> **Q3**: In the online learning setting, we may have no information about the state of other agents. In such cases, what can we do?
>
> **A**: Our paper considers the centralized online learning setting, meaning that there exists a central controller for moderating all the agents. When there is no information about the state of other agents, we may resort to the “belief state” for representing the state of other agents. We can also consider the decentralized online MARL problem, yet it is beyond the scope of the current paper.
>
> **Q4**: For eq.5, there is an i.i.d assumption for each kernel mean embedding $\omega$, which could be problematic.
>
> **A**: The i.i.d. assumption of each kernel mean embedding at a fixed time step is reasonable, since under a fixed behavior policy, we can collect i.i.d. sample trajectories starting from a random initial state. Moreover, we can relax the i.i.d. assumption by using martingale techniques to analyze the uncertainty in estimating the Bellman operator. We leave it as a future direction.

---

> > ### Comment · Reviewer_WSQ9 · 2021-08-27
> > **a nice theoretical contribution but no any experimental verification**
> >
> > Thanks for the authors' explanation and based on that,  I would increase my score to 6. Overall, I think that this is a nice theoretical contribution to the community of MARL. However, I'm not completely satisfying because I did not  see how the analytical properties of the proposed algorithm make their way to practical performance, which is equally important as well.

---

> > > ### Author Response · Authors · 2021-08-27
> > > **Thank you for your encouraging comment**
> > >
> > > Thank you for your encouraging comment. We appreciate your valuable suggestions on bridging the theory to practice. We will empirically verify our technical assumptions and evaluate the performance of proposed algorithms.

---

### Official Review · Reviewer_yJWX · 2021-07-25

**Rating:** 6
**Confidence:** 3

**Summary:**

The paper introduces an offline algorithm named SAFARI for solving Mean Field Games, using RKHS embedding of the infinite dimensional distribution of states. No experience are provided, but a bound on the efficiency of the method in terms of approximated value function is provided. This bound depends on the time horizon, the number of offline available trajectories as well as the effective dimension of the for parametrising the value function. It is valid whenever the set of available trajectories satisfies a natural weak coverage assumption. As a by-product, the authors also provide an online version of the algorithm.

**Limitations And Societal Impact:**

I may have missed it but did not clearly identify where the limitations of the proposed approach are presented in the paper.

**Main Review:**

Main comments:
1. The offline learning of Nash equilibria in Mean Field Games is a relevant and new topic, up to my knowledge. The distribution embedding using RKHS approach seems rigorously treated and well chosen for this setting. The literature review is complete and accurate and the paper is overall well written and easy to read.
2. Can you justify why your method is indeed approximating the Nash of an MFG and not the solution of a corresponding Mean Field Control problem? As you mainly focus on the underpinned value function, this may need clarification.
3. You are using a metric indicating how close the induced approximated value function is from the optimal one, how does it translates in terms of quality of Nash (with a metric such as exploitability for example) or approximation of the state distribution, which is also a commonly used metric for evaluating the quality of a MFG solution.
4. The class of MFG considered here is still unclear for me. The abstract mentions an episodic MFG while the main paper is less clear about that point. Which assumptions are present in your setting on the dynamics and reward structure, ensuring the existence and more importantly the uniqueness of the Nash?
5. When using a Mean Field setting for solving MARL problems, one naturally  in an offline setting accumulates trajectories from the N-agents system,  which itself is an approximation of the MF limit. This probably goes beyond the scope of this paper but do these  additional errors fit in your approach?
6. The theoretical results are interesting (I did not check everything) but could have been completed by experimental results, even in small dimensional settings. This would have been a good occasion to compare online and offline efficiency.

Typo: l 128 "an representative";

**Time Spent Reviewing:**

2 hours

---

> ### Author Response · Authors · 2021-08-10
> **Author response to Reviewer yJWX**
>
> Thank you for your comments.
>
> **Q1**: (Major comment 1) The offline learning of Nash equilibria in Mean Field Games is a relevant and new topic, up to my knowledge.
>
> **A**: We would like to restate the difference between the Mean-Field MARL (MF-MARL) problem considered in the paper and Mean-Field Games (MFG). Specifically, our MF-MARL problem studies the collaborative setting. The agents are homogeneous, and the goal is to search for an optimal (global) policy for the representative agent. Such a policy is also to be deployed to all the homogeneous agents. In sharp contrast, MFGs focus on finding optimal local policies for competitive agents, which is further termed as the **(mean-field) Nash equilibrium**. Such an equilibrium, however, is not studied in our MF-MARL problem. Nor do our proposed algorithms aim to find Nash equilibria.
>
> Furthermore, it might be easier to see the difference between MF-MARL and MFG in the finite-agent setting. When the number of agents is finite, MF-MARL aims to find the policy that maximizes the average returns of all agents, assuming the agents are permutation invariant. The average (sum of) return of all agents is also known as the utilitarian social welfare function and the optimal policy maximizes such a notion of social welfare. Whereas in a finite-agent MFG, the optimal policy corresponds to the classical Nash equilibrium policy that satisfies permutation invariance, i.e., the policy, given that all other agents adopt it, maximizes the return of a selfish agent.
>
> **Q2**: (Major comment 2) Can you justify why your method is indeed approximating the Nash of an MFG and not the solution of a corresponding Mean Field Control problem? As you mainly focus on the underpinned value function, this may need clarification.
>
> **A**: We consider the MF-MARL problem, which is different from MFG, and therefore, our proposed offline algorithm does not approximate the Nash of an MFG.
>
> On the other hand, solving the optimal policy in our MF-MARL problem can be viewed as solving a discrete-time mean-field control problem. To be more specific, in the $N$-agent setting, our formulation of MF-MARL corresponds to the following episodic mean-field control problem:
> Let $[N] = \{1, \dots , N\}$ be the $N$-agents and let $H$ be the horizon. Each agent has a local state $x_h^i \in \mathcal{X}$ and local action $a_h^i \in \mathcal{A}$ for all $i\in[H]$ and $h \in [H]$. At the $h$-th time step, each agent observes its local state $x_h^i$, and the current mean-field state $d_h $, which is the empirical measure of $\{ x_h^i, i\in [N]\}$, and takes a local action $a_h^i$. Then, agent $i$ receives an immediate reward given by $r_h (x_h^i, d_h, a_h^i)$. Meanwhile, the local state transits to $x_{h+1}^i$ which is sampled from $p_h(\cdot | x_h^i, d_h, a_h^i)$.
>
> The goal is to find a local policy which maps $(x_h^i, d_h)$ to $a_h^i$ such that, when every agent adopts such a policy, the average value of the total return of all the agents, $\frac{1}{N} \mathbb{E} [ \sum_{h=1}^H \sum_{i=1}^i r_h^i ],$  is maximized.
>
> Notice that the local reward function and the transition function of the local state are the same for each agent. Moreover, they depend on the information about other agents only via the mean-field state. Thus, in this problem, the agents are permutation invariant. Our MF-MARL corresponds to the limiting regime where $N$ goes to infinity. In this setting, $\frac{1}{N} \mathbb{E} [ \sum_{h=1}^H \sum_{i=1}^i r_h^i | x_h^i = s]$ converges to the value function $V_1^{\pi}$ of the representative agent.
> We will add the connection to mean-field control in the next version.
>
>
> **Q3**: (Major comment 3) You are using a metric indicating how close the induced approximated value function is from the optimal one, how does it translates in terms of quality of Nash (with a metric such as exploitability for example) or approximation of the state distribution, which is also a commonly used metric for evaluating the quality of a MFG solution.
>
> **A**: In our MF-MARL problem, $V_1^{\pi}$ is the expected return obtained by the representative agent when adopting policy $\pi$, which is also equal to the average return of all agents when all agents use $\pi$. Thus, $V_1^{\pi^*}$ is the optimal value of the average return among all agents, i.e., the optimal value of utilitarian social welfare. Therefore, since the goal of MF-MARL, finding the socially optimal policy, is exactly to maximize $V_1^{\pi}$, the sub-optimality gap defined in Line 247 characterizes how far $\hat \pi$ is from the optimal policy.
> However, since our focus is finding the policy that maximizes the social welfare, instead of the Nash policy that maximizes the return of a selfish individual agent, having a small sub-optimality does not imply any proximity to the Nash equilibrium.
>
> **Q4**: (Major comment 4) The class of MFG considered here is still unclear for me. The abstract mentions an episodic MFG while the main paper is less clear about that point. Which assumptions are present in your setting on the dynamics and reward structure, ensuring the existence and more importantly the uniqueness of the Nash?
>
> **A**: Our paper studies episodic MF-MARL problems rather than episodic MFG (in fact, we did not mention episodic MFG in the abstract).
>
> We provide a detailed description of the MF-MARL problem in Section 3. In more detail, the reward function and the transition probabilities are linearly parameterized in an RKHS by equation (4). We further provide regularity assumptions on the kernel spectrum (Assumption 2) and “weight” parameters $v_h$ and $\theta_h$ (Assumption 3). In addition, we do not need the existence and uniqueness of the Nash in our theory.
>
> **Q5**: (Major comment 5) When using a Mean Field setting for solving MARL problems, one naturally in an offline setting accumulates trajectories from the N-agents system, which itself is an approximation of the MF limit. This probably goes beyond the scope of this paper but do these additional errors fit in your approach?
>
> **A**: Our analysis has already handled the finite agent approximation error of the mean-field limit. The high level idea is depicted in the technical overview part in Section 5, where the uncertainty quantifier $\Gamma_h$ of estimating the Bellmen operator consists of the covariate concentration error on mean embedding due to finite agents (Line 296). The detailed analysis can be found in Lemma 3 and Lemma 4 in the supplementary, which establishes an $1/\sqrt{m}$-rate concentration of the mean embedding.
>
> **Q6**: (Major comment 6) The theoretical results are interesting (I did not check everything) but could have been completed by experimental results, even in small dimensional settings. This would have been a good occasion to compare online and offline efficiency.
>
> **A**: We will provide some experiments in the next version.

---

> > ### Comment · Reviewer_yJWX · 2021-08-27
> > **Thanks !**
> >
> > Thanks you for precisely answering my concerns.
> > Looking forward to see experiments in the final version and I'm happy to move my rating up to 6.

---

### Decision · Program_Chairs · 2021-09-27

**Decision:**

Accept (Poster)

**Comment:**

The reviewers raised some concerns with regard to lack of experimental results in this paper. Despite that they agree that the paper considers an interesting and relevant problem and it provides non-trivial theoretical results in terms of sample complexity of the proposed pessimistic method. Please provide the simple illustrative experiments in the final version.